# Conditional Generative Modeling for De Novo Hierarchical Multi-Label Functional Protein Design

## Abstract

The availability of vast protein sequence information and rich functional annotations thereof has a large potential for protein design applications in biomedicine and synthetic biology. To this date, there exists no method for the general-purpose design of proteins without any prior knowledge about the protein of interest, such as costly and rare structure information or seed sequence fragments. However, the Gene Ontology (GO) database provides information about the hierarchical organisation of protein functions, and thus could inform generative models about the underlying complex sequence-function relationships, replacing the need for structural data. We therefore propose to use conditional generative adversarial networks (cGANs) on the task of fast *de novo* hierarchical multi-label protein design. We generate protein sequences exhibiting properties of a large set of molecular functions extracted from the GO database, using a single model and without any prior information. We shed light on efficient conditioning mechanisms and adapted network architectures thanks to a thorough hyperparameter selection process and analysis. We further provide statistically- and biologically-driven evaluation measures for generative models in the context of protein design to assess the quality of the generated sequences and facilitate progress in the field. We show that our proposed model, ProteoGAN, outperforms several baselines when designing proteins given a functional label and generates well-formed sequences.

## 1 Introduction

Designing proteins with a target biological function is an important task in biotechnology with high-impact implications in pharmaceutical research, such as in drug design or synthetic biology (Huang et al., 2016). However, the task is challenging since the sequence-structure-function relationship of proteins is extremely complex and not yet understood (Dill & MacCallum, 2012). Functional protein design is currently done by traditional methods such as directed evolution (Arnold, 1998), which rely on a few random mutations of known proteins and selective pressure to explore a space of related proteins. However, this process can be time-consuming and cost-intensive, and most often only explores a small part of the sequence space. In parallel, data characterizing proteins and their functions is readily available and constitutes a promising opportunity for machine learning applications in protein sequence design. Moreover, the hierarchical organisation of protein functions in a complex ontology of labels could help machine learning models capture sequence-information relationships adequately. Recently, generative models have attempted to design proteins for different tasks, such as developing new therapies (Muller et al., 2018; Davidsen et al., 2019) or enzymes (Repecka et al., 2019). Nonetheless, most of the *de novo* protein sequence design methods, which generate sequences from scratch, focus on a specific function or on families of short proteins. Instead, we would like to focus on modeling several different biological functions at the same time to eventually be able to freely combine them. To this end, one first requires a model that is able to deal with and to understand the inherent label structure. We concern ourselves with the development of such a generative model.

In this work, we introduce the general-purpose generative model ProteoGAN, a conditional generative adversarial network (cGAN) that is able to generate protein sequences given a large set of functions in the Gene Ontology (GO) *Molecular Function* directed acyclic graph (DAG) (Gene On-

tology Consortium, 2019). To the extent of our knowledge, we are the first to propose a hierarchical multi-label *de novo* protein design framework, which does not require prior knowledge about the protein, such as seed sequence fragments or structure.

Our contributions can be summarized as follows: (i) we propose a data-driven approach to *de novo* functional protein generation that leverages a large set of annotated sequences, (ii) we present a new extensive evaluation scheme to assess validity, conditional consistency, diversity, and biological relevance of the generated sequences, and (iii) we conduct an in-depth model optimization to derive actionable insights on architectural choices and efficient conditioning mechanisms while outperforming existing state-of-the-art protein generators.

We focus on generative adversarial networks, due to their promising performance on specific sequence design tasks (Repecka et al., 2019). We choose a conditional setting not to rely on oracles nor on multiple rounds of training-generation-measurement, since to this date a well performing general-purpose predictor of protein function remains elusive (Zhou et al., 2019). As opposed to most existing methods (see Section 2), we aim to generate a comprehensive variety of proteins exhibiting a wide range of functions, rather than focusing on optimising a single function within a unique protein family. As this is a different task from the ones found in the literature, we need to define an adequate evaluation pipeline.

Therefore, we establish a multiclass protein generation evaluation scheme centered around validity and conditional consistency. The model should generate protein sequences whose distribution resembles that of natural proteins and hence have similar chemo-physical properties, and it should do so *conditionally*, namely generating proteins of a given functional class without off-target functions.

We are hence confronted with the problem of assessing i) the performance of the generative model in a general sense, which is defined by how well the generated distribution fits the training data distribution, and ii) the conditional performance of the model which we define as a special case of the general performance, where we compare sequence feature distributions between labels. We therefore require distribution-based evaluations. A natural choice to evaluate the performance of a generative model is a two-sample test, which allows to answer whether a generated and a real set of samples (i.e. the dataset) could originate from the same distribution. The difficulty here is to define a measure that can handle the structured data, in our case protein sequences. To this end, we design Maximum Mean Discrepancy (MMD)-based evaluation criteria (Gretton et al., 2012), which ensure good model performance and a functioning conditioning mechanism by measuring differences in empirical distribution between sets of generated and real protein sequences. To ensure diversity, we monitor the duality gap (Grnarova et al., 2019), a domain-agnostic indicator for GAN training. Lastly, we use a series of biologically-driven criteria in the evaluation phase that confirms the biological validity of the generated protein by relying on the standard protein feature software ProFET (Ofer & Linial, 2015).

With this arsenal of measures, and given the low computational complexity of our MMD-based criteria, we compare different architectural choices and hyperparameters in an extensive and efficient Bayesian Optimization and HyperBand (BOHB) (Falkner et al., 2018) search. In particular, we develop improved variants of two existing conditional mechanisms on GANs (Odena et al., 2017; Miyato & Koyama, 2018) and show for the first time that the previously unexplored combination of both is beneficial to conditional generation. Moreover, the selected model outperforms (i) *de novo* conditional model CVAE (Greener et al., 2018), repurposed and trained towards functional protein generation, other introduced baselines (HMM, n-gram model), and (ii) models specifically built to challenge the necessity of a conditional mechanism.

The remainder of the document is organized as follows. First, the background and related work section gives a concise overview of the biological mechanisms underlying the function of proteins, summarises the state-of-the-art generative models applied to protein design, details some conditional mechanisms in GANs and identifies existing evaluation criteria for GANs and cGANs. Subsequently, the method section describes ProteoGAN and its components and explains our protein generation evaluation framework. Finally, the results obtained by conditioning the generation of new sequences on 50 GO classes are presented and discussed before concluding with some final remarks.

## 2 BACKGROUND AND RELATED WORK

**Biological mechanisms underlying protein functions.** Proteins are biological structures that serve a wide variety of purposes in organisms. They are composed of chains of amino acids and can therefore be represented as simple sequences. However, the relationship between physico-chemical properties of amino-acids, three dimensional structure and resulting biological activity of the macro-molecule is highly complex (see supplementary Section A.1). Nevertheless, since the advent of modern sequencing techniques, millions of proteins have been registered in databases, along with curated descriptions of their function. For example, the GO is a species-agnostic ontology that aims at classifying genes (and the resulting proteins) according to their functions, locations, and governing biological processes using a hierarchical structure of functional labels. As such, it represents an ideal interface between scientists who wish to design proteins with descriptive and modular labels, and a generative model that captures the complex relationships of sequence, structure and function.

**Guided and conditional generative models.** Machine learning models and more recently deep generative models (Eddy, 2004; Goodfellow et al., 2014; Kingma & Welling, 2014; Rezende et al., 2014; Vaswani et al., 2017; Li et al., 2017a) have been used to design *in silico* biological sequences, such as RNA, DNA or protein sequences (R. Durbin & Mitchinson, 1998; Davidsen et al., 2019; Brookes et al., 2019; Hawkins-Hooker et al., 2020; Costello & Martin, 2019; Anand & Huang, 2018). Among them, several approaches have been developed in order to control sequence generation. They can be sorted in three categories, *guided*, *conditional* or combinations thereof. Guided approaches use a predictor oracle in order to guide the design towards target properties, through iterative training, generation and prediction steps (Brookes et al., 2019; Gane et al., 2019; Angermueller et al., 2019; Gupta & Zou, 2019; Killoran et al., 2017; Repecka et al., 2019). While these guided methods have the theoretical advantage to produce proteins with specific characteristics, for example brightness (Brookes et al., 2019), they require an independent oracle. This oracle can be itself hard to train and remains imperfect, even for highly specialized prediction tasks. Moreover, the lack of well-functioning predictors for large numbers of labels impairs the usage of guided-generation techniques to multiclass applications such as functional protein generation (Zhou et al., 2019). On the contrary, conditional approaches integrate the desired properties in the generation mechanism, eliminating the need for an oracle. Karimi et al. (2019) provided a guided conditional Wasserstein-GAN to generate proteins with novel folds. Interestingly, Madani et al. (2020) developed ProGen, a conditional transformer that enables a controlled generation of a large range of functional proteins. However, the method's need for sequence context can be experimentally constraining and is not compatible with *de novo* design. Ingraham et al. (2019) present a graph-based conditional generative model that unfortunately needs only sparsely available structural information. Das et al. (2018) and Greener et al. (2018) train conditional VAEs in order to generate specific proteins, such as metalloproteins.

**Conditional mechanisms in GANs.** Several conditional mechanisms have been proposed to conditionally generate samples with GANs. Among the most successful ones in conditional image generation tasks, Odena et al. (2017) introduced the auxiliary classifier GAN (AC-GAN), which uses a third integrated network, in addition to the generator and the discriminator, to predict labels of both real and generated inputs to the discriminator. Miyato & Koyama (2018) proposed an alternative conditioning mechanism, where the label information is introduced to the network as the inner product of the embedded label vector and an intermediate layer of the network, a mechanism they refer to as projection. Projections can be seen as an alternative to simple concatenations of label information to the network input (Mirza & Osindero, 2014), in a way that respects the underlying probabilistic model.

**Generative models evaluation.** To this date, there is no definitive consensus on the best evaluation measures for the evaluation of *quality*, *diversity* and *conditional consistency* of the output of a (conditional) generative model (Papineni et al., 2002; Salimans et al., 2016; Heusel et al., 2017; Shmelkov et al., 2018; Kynkäänniemi et al., 2019; DeVries et al., 2019). Most measures that stand out in computer vision such as the Inception Score (IS) (Salimans et al., 2016), the Frechet Inception Distance (FID) (Heusel et al., 2017), GAN-train and GAN-test (Shmelkov et al., 2018) depend on an external, domain-specific predictor. On the contrary, the domain-agnostic duality gap can be computed during training and at test time, and has been shown to correlate well with FID (Grnarova et al., 2019). In functional protein prediction, results obtained by state-of-the-art classification mod-

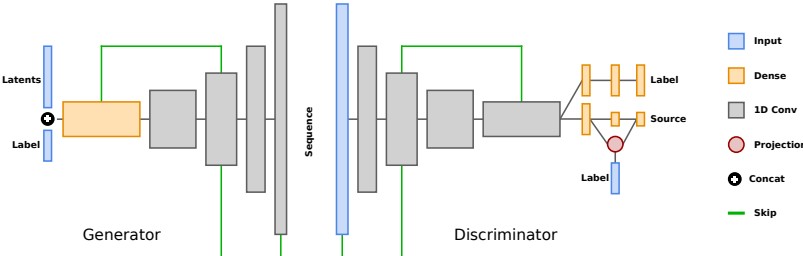

Figure 1: A schematic overview of the general model structure that we screened during hyperparameter search. Architectural features such as layer number and skip connections can vary. We note that the architecture of our final model differs from the one depicted here.

els are encouraging but still not good nor fast enough to entirely rely on them when evaluating and training GANs ($F_{max} = 0.631$ (Radivojac et al., 2013; Zhou et al., 2019; You et al., 2019).

## 3 METHODS

### 3.1 MODEL ARCHITECTURE

We chose the framework of Generative Adversarial Networks for our model, specifically the Wasserstein-GAN with Gradient Penalty (Arjovsky et al., 2017; Gulrajani et al., 2017). Our convolutional architecture resembles the funnel-like structure of DCGAN (Radford et al., 2015). We propose variants of existing models in order to adapt the framework to sequence generation and to guide future model development in the field. An extensive parameter search is performed on a validation set to select the best variants and hyperparameters (see Section 4) while a schematic view of the model can be found in Figure 1.

**Conditioning mechanism.** We allow for the insertion of three types of conditioning mechanisms: projections, auxiliary classifiers, or a combination of both. While projection or auxiliary classifiers are not exclusive, we did not encounter any work that used both in one model. The conditioning mechanisms are further explained in the supplementary Section A.3.1. We also allow for more than one projection from different layers of the discriminator, with the rationale that in this way we could utilize protein sequence information of the different abstraction levels of the convolutional layers. Finally, the embedded label is always concatenated to the latent noise vector input of the generator.

**Hierarchical label encoding.** Given the hierarchical structure of the functional labels, we allow for three types of label embeddings: a) one-hot encoding, as a common encoding for labels, b) Poincaré encoding (O'neill, 2014), as the hyperbolic space is well-suited for hierarchical data and c) node2vec encoding (Grover & Leskovec, 2016), which preserves neighbourhood relationships by encoding the nodes of the GO DAG based on random walks. One protein can present several GO labels, and these embeddings aim to capture the relations between the labels in the DAG and to incorporate this information into the generative process. We compare against a baseline model without this hierarchical information (named *Non-Hierarchical*), where sequences are fed to the model for each label independently.

We further allow to concatenate chemophysical properties of the respective amino-acids to the one-hot encoding of the sequences. Other architectural and optimizer hyperparameters are subject to optimization, whose descriptions and value ranges are detailed in the supplementary Section A.2.3.

### 3.2 MODEL EVALUATION

#### 3.2.1 MMD TO MEASURE SEQUENCE VALIDITY AND CONDITIONAL CONSISTENCY

**Computation of MMD.** We propose to use the kernel two-sample test Maximum Mean Discrepancy (MMD) (Gretton et al., 2012; Sutherland et al., 2016) to build evaluation criteria for conditional sequence design. The test has been shown to be suited for distinguishing distributions of structured data such as protein sequences (Borgwardt et al., 2006). MMD has also been explored in the context of GANs, where it was shown to be able to function as a discriminator (Li et al., 2015; Dziugaite et al., 2015). Here we use it to assess model quality based on samples. Let $X = \{x_i\}_{i=1}^n$ and

$Y = \{y_j\}_{j=1}^m$ be samples from the distributions of generated and real proteins sequences, respectively $P_g$ and $P_r$. When using MMD, we compute the empirical squared MMD statistic between these two distributions using equation (2) of Gretton et al. (2012):

$$\text{MMD}^2[\mathcal{F}, X, Y] = \left\| \frac{1}{n} \sum_{i=1}^n \frac{\phi(x_i)}{\|\phi(x_i)\|_2} - \frac{1}{m} \sum_{j=1}^m \frac{\phi(y_j)}{\|\phi(y_j)\|_2} \right\|_2^2 \tag{1}$$

where $\phi(\cdot) \in \mathcal{F}$ is the variant of the mapping function of the spectrum kernel proposed by Leslie et al. (2001), which accounts for the presence or absence of k-mers in the sequences of interest. The size of the k-mers was set to 3, as suggested for protein sequences by Leslie et al. (2001). This expression is fast to compute as it scales linearly with the number of sequences and can consequently be used as an early stopping criterion during the training process and for model selection. We also report the result of a more powerful Gaussian kernel on top of the 3-mer embeddings. We used the statistic itself rather than the resulting p-values because these latter are too sensitive as soon as more than 3% of random noise is added to the sequences (Table A6 in the Supplementary Material).

**Generation of *in-silico* validated sequences with MMD.** In order to assess to which extent our model captures the unconditional distribution of our sequences, we use MMD as described above. We therefore first ensure that the proteins generated by the model resemble existing ones. We could show that a 3-mer embedding is sufficient for our context, as the functional classes we are concerned with (see Section 4 for description of functional classes) can be linearly separated in feature space. The functional annotations can be classified with 94% accuracy based on hyperplanes in our embedding space in a one-vs-all scheme.

**Generation of functional sequences with MRR on conditional MMD.** We compute the conditional performance by measuring, for each set of generated proteins for a given target label, how many sets of real proteins with an off-target label are closer than the set of real proteins of the targeted label. The distances between sets are estimated using MMD as defined above. Let $R$ be a set of real sequences, which is composed of the sets $\{R_i\}_{i=1}^d$ of sequences with annotated labels $\{L_i\}_{i=1}^d$, where $d$ is the number of labels. Let $G = \{G_i\}_{i=1}^d$ be an equally structured set of generated sequences. We want to maximise the following mean reciprocal rank (MRR):

$$\text{MRR}(R, G) = \frac{1}{d} \sum_{i=1}^d \frac{1}{\text{rank}(\text{MMD}(R_i, G_i))} \tag{2}$$

where $\text{rank}(\text{MMD}(R_i, G_i))$ is the rank of $\text{MMD}(R_i, G_i)$ among sorted elements of the list $[\text{MMD}(R_i, G_1), \text{MMD}(R_i, G_2), \ldots, \text{MMD}(R_i, G_d)]$. $\text{MRR}(G)$ is maximal and of value 1 when the generated sets of proteins for a given label are the closest to the set of real proteins with the same label. Variants of MRR that give more insight on conditional performance for closely-related functions in the label hierarchy are described in the supplementary Sections A.3.2 and A.3.3.

### 3.2.2 MEASURES TO ASSESS QUALITY AND DIVERSITY OF GENERATED SEQUENCES

We monitor the duality gap (Grnarova et al., 2019) of our GAN model. A small duality gap indicates good convergence and common failure modes, such as mode collapse, can be detected. We follow the authors' suggestion to approximate the latter with past snapshots of the training. Additionally, to provide a protein-centric evaluation we also report Kolmogorov-Smirnoff (KS) two sample tests (Massey, 1951) between generated and real samples from the $\sim 500$ (not k-mer related) sequence-based features from the feature extractor library ProFET (Ofer & Linial, 2015).

## 4 EXPERIMENTAL SETUP

**Data.** Sequence data was acquired from the UniProt Knowledgebase (UniProtKB, Consortium (2019)). Out of the 180 million entries, we selected sequences with experimental evidence and at least one annotated GO label. Then we restricted to the standard amino acids and a sequence-length of 2,048, which covers ca. 98.5% of the remaining data points. The resulting dataset contains 149,390 sequences. The ontology is structured as a DAG and labels have a hierarchical relationship,

i.e. proteins with a given functional label inherit automatically the labels of their parents in the DAG. We restricted the number of labels to the 50 most common molecular functions, imposing a threshold of at least 5,375 sequences per functional label. This is sufficient for a proof-of-principle and would even enable the design of experimental assays for validation. Figure 4 illustrates the selected subset of labels and their hierarchical relationships. We randomly split the dataset in training, validation and test sets keeping ca. 15,000 (10%) sequences each in the validation and test sets. During model optimization, we use smaller splits with ca. 3.000 sequences each. We ensure that all labels have a minimum amount of samples in the test and validation sets, and use the same number of sequences per class for our MRR measure (1.300 and 300 sequences, respectively). Further details about the dataset and splits are available in the supplementary Section A.2.1 and Figure A1.

**Comparison partners.** We compare ProteoGAN to several baselines to put its performance into perspective. We first use CVAE (Greener et al., 2018). We performed a bayesian optimization hyperparameter search over 1,000 models. The model was adjusted to incorporate the 50 labels of our problem setting. We could not compare fairly to PepCVAE (Das et al., 2018) as the model does not scale to sequence lengths of 2048 amino-acids. We refer the reader to the respective papers and to Section A.2.2, Tables A2-A3 and Figure A2 for further information and results on both baselines.

Additionally, we constructed several baselines with the goal to assess the usefulness of the conditional generation mechanism. The first baseline, referred to as *Unconditional*, consists of as many unconditional GANs as there are labels. To do so, we remove the conditioning mechanism from our model and train multiple instances on the fifty subsets of data that are annotated with a particular label. We generate sequences for a target label by sampling them from the GAN trained on the sequences annotated with the same label. Our second alternative to conditioning replaces a conditional model with the combination of an unconditional model trained on the full data and an established predictor of protein function, NetGO (You et al., 2019), used to predict the labels of generated sequences which replaces the need for conditioning. We refer to this baseline as *Predictor-Guided*. Further, we assess whether the model utilizes the hierarchical label structure by training a *Non-Hierarchical* baseline which only sees the labels independently for each sequence. This is done by replicating and annotating a sequence for each associated label, while keeping the number of gradient updates the same across all models.

Lastly, we compare two traditional generative models for sequences, a (profile) HMM and an n-gram model (n=3). Since they are without conditioning mechanism, we train one model for each label combination in our testset (called 'on label set' in Table 1, in this case a protein's GO labels can contain several GO terms and their parent terms), as well as one model for each of the 50 labels we are dealing with (called 'on single label' in Table 1, in this case we discard the hierarchy of labels). Further description of most of the baselines are available in the Supplementary Material Section A.2.2.

**Hyperparameter optimization.** We conducted two Bayesian Optimization and HyperBand (BOHB) searches (Falkner et al., 2018) on six Nvidia GeForce GTX 1080, first a broad search among 23 hyperparamaters and a second, smaller and more selective, among 9 selected hyperparameters. The optimization objective was set to maximize the ratio of the evaluation measures MRR/MMD, which are detailed in Section 3.2.1, to balance the validity and the conditional consistency of the generated sequences. Both searches were complemented by a functional analysis of variance (fANOVA) (Hutter et al., 2014). The results of the second optimization, for which we evaluated $1,000$ models for a maximum duration of 27 epochs in our experiments, are shown Figure 2. The 27 best selected models of the second hyperparameter search were then trained for a prolonged duration of 100 epochs, where the conditioning mechanism and an associated weighing factor became most important. Further details about hyperparameter optimization are available in the supplementary Section A.2.3 and Table A4.

## 5 RESULTS AND DISCUSSION

### 5.1 MODELS SELECTED BY THE BOHB OPTIMIZATION

**Insights on cGANs architecture.** The results of the fANOVA and of a prediction of hyperparameter marginals on the second automatic hyperparameter optimization led to the following observations. We could show that adding chemophysical features did generally decrease model performance, and

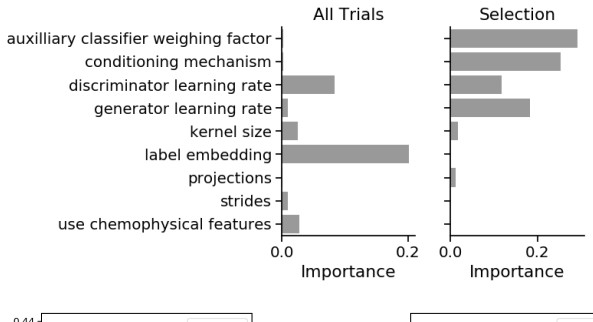

Figure 2: Hyperparameter importance. The bars show individual importance of each hyperparameter in terms of the variance they explain. We conducted the analysis for all trials of the optimization (left) and for the selected models that were trained for prolonged time (right). The total variance explained by the main effects was 36% and 88%, respectively.

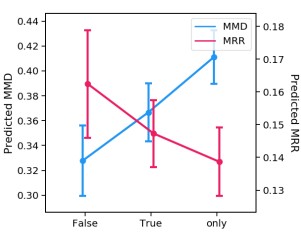
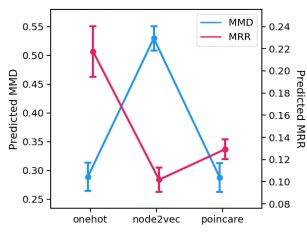
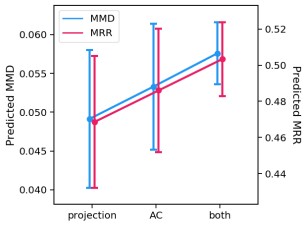

(a) Use biochemical features     (b) Label embedding     (c) Conditioning mechanism

Figure 3: Marginal predictions of biochemical features and the label embedding based on optimization data and of the conditional mechanism based on data of the 27 best selected models. Predictions were obtained training on MMD and MRR. Lower is better for blue and higher is better for red.

that the most suitable label embedding is a simple one-hot encoding (Figure 3a-b). More interestingly, the conditioning mechanism showed different impacts when analysing with respect to either MMD and MRR (Figure 3c). Performance increased when changing from projection, to auxiliary classifier, to both when evaluating with respect to MRR, but the opposite occurred for MMD. This indicates that there is a trade-off between conditional generation and general sequence quality. A combination of both conditioning mechanisms is the best option when aiming at conditional performance, although others might prefer an emphasis towards better sequence quality over conditional generation. We further conclude that small and simple model architectures show best results from an analysis of our first optimization, and that many of the various proposed extensions to the GAN framework did not show significant impact on performance in our context (see supplementary Figures A3, A4, A5, A6).

**Selection of the final model.** We selected the best model checkpoint over a training phase of 300 epochs based on the validation set. The final hyperparameter configuration of our model (ProteoGAN), as well as loss plots and real-time evaluations during training can be found in supplementary Section A.3.4, Table A5, and Figure A7. Most notably, the final model had both conditioning mechanisms, multiple projections, and one-hot encoding of label information. Additionally, the duality gap evaluations during training (Figure A7) showed no signs of mode collapse.

## 5.2 Performance evaluation of ProteoGAN

We report measures on the test set of the best model in Table A8. We additionally report the model performance per individual GO label in Figure 4. ProteoGAN can reliably generate sequences conditioned on various GO labels, where many of the labels can be very well generated without major off-target effects (23 (resp. 32) of 50 labels were on average ranked first or second (resp. or third) compared to all other labels). The overall conditional performance (MRR=0.545) is reasonably

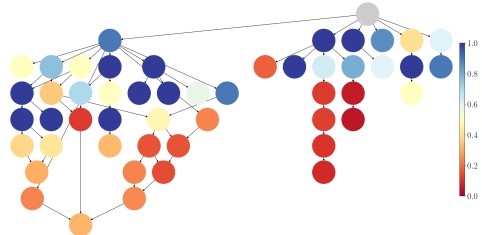

Figure 4: Reciprocal rank for each individual label. The structure represents the relations in the GO DAG. Nodes are colored by how well the model can target them. Blue to light orange indicates that the model ranks the target function in the first three positions in average. Values are averaged over $n = 5$ different data splits.

close to a reference set of natural protein sequences (MRR=0.889). With respect to general sequence quality, ProteoGAN reaches MMD values of 0.040, which corresponds to roughly 20% of random mutations in a set of natural sequences (compare supplementary Table A6). While this number is high compared to traditional protein engineering approaches, we note that systematic studies showed that proteins can tolerate a high mutation rate (up to 59% (Markiewicz et al., 1994; Ng & Henikoff, 2001)) as long as critical residues are not affected, and notably Repecka et al. (2019) could experimentally validate GAN-generated sequences with up to 34% difference to its closest homolog. Additionally, we show that the sequences generated by the model are not closer to the training set than the test set is, as measured by squared euclidean distance in kernel feature space (Supplementary Figure A8). This implies that our model is not overfitting on the training set and is able to generate novel sequences.

Table 1 shows the performance under MMD, Gaussian MMD and MRR for ProteoGAN and various baselines. In general, MMD and Gaussian MMD give similar rankings for the different models. We first note that ProteoGAN outperforms all multi-label conditional baselines by a significant margin in both overall model performance and conditional generation. We could also show that the (conditional) ProteoGAN is comparable to the *Unconditional* model, which implies that the conditioning mechanism can substitute the training of many models per individual label. It shows that the conditioning mechanism is working well. ProteoGAN scores are also better than the *Non-Hierarchical* model, which shows that it could incorporate the hierarchical multi-label structure into the generation and that it is beneficial. It remains to be shown that this is sufficient to enable out of distribution functional label generation.

The weak conditional performance of the *Predictor-Guided* model (MRR = 0.109) suggests that the state-of-the-art predictor used (NetGO) is not able to predict the right label for the generated sequences, and therefore fails at guiding conditional generation, possibly because the generated sequences do not have close homologs in the protein databases.

As an outlook, we provide some results of ProteoGAN trained on more specific labels (lower panel in Table 1). We kept the same architecture as the one optimised for 50 labels and retrained the model in three different situations. We observe that the performance is still reasonable when the number of labels is doubled (100 labels). With 200 labels the performance starts to drop. It may be that the model capacity is too low in this case, which could be alleviated by tuning the hyperparameters we have identified to be critical. When training the model on the more specific labels of the *depth-first* sampling of labels the performance stays good, however we note that with increasing specificity the classes get very small.

ProteoGAN further outperformed CVAE and the HMM and n-gram baselines. This becomes evident especially in conditional generation (MRR). Figure 5 confirms these results. It can generally be seen that ProteoGAN shows good agreement with the real protein sequences of the testset in several important biological sequence features. We show exemplary feature distributions for some of the ca. 500 features analyzed by ProFET. For a summary statistic of all features, we report the distribution of KS-statistics between generated and real data across all features. Also here, the ProteoGAN variants outperformed the other models.

## 6 CONCLUSION

In this work, we develop a conditional generative adversarial model, ProteoGAN, which generates sequences validated *in-silico* according to statistically- and biologically-driven measures. We identify useful architectures and extensions of cGANs for proteins. The measures we propose are fast to compute and therefore can be used during optimization and training, as well as for model assessment. We show that the conditioning on hierarchical Gene Ontology labels was successful, as we could show that a number of labels can be well targeted. Generally, it remains to be shown that the class of multi-label generative models can not only generate the correct feature distributions, but also experimentally valid proteins with the specified functions. This requires further development of evaluation measures, which we hope to have set a basis for. Future improvements to the model might also come from a larger number of labels, more specific targeting for small classes and a proof that such conditional models are able to combine the modular GO labels into new and unseen functions, which would be tremendously useful for biotechnological applications.

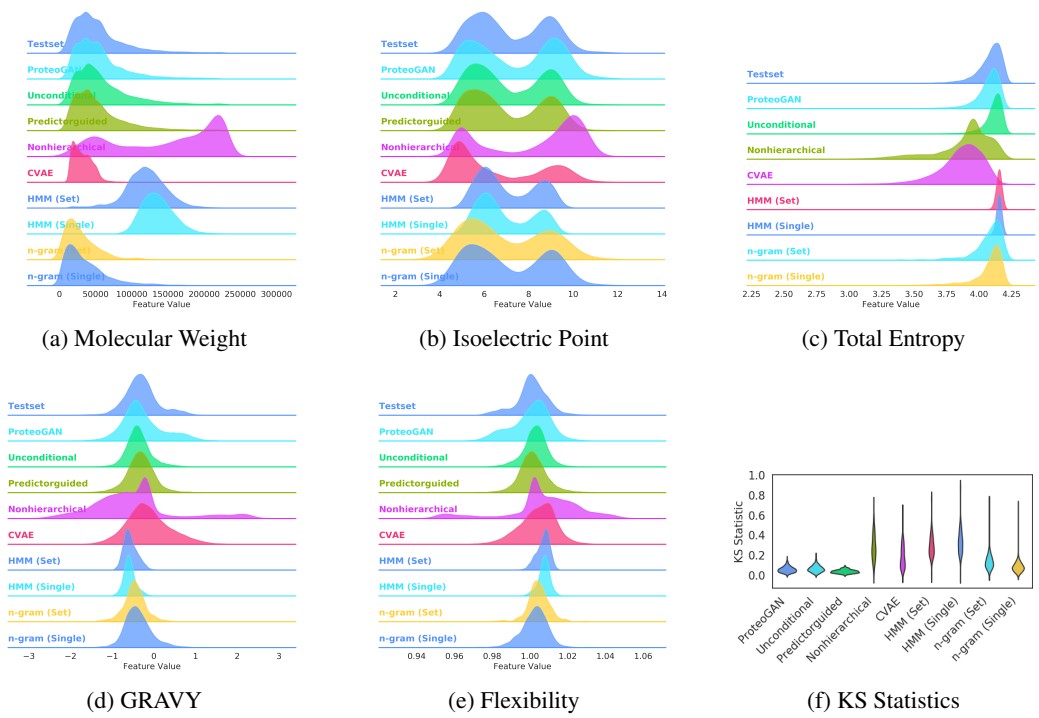

Figure 5: (a-e) Distributions of some selected features of ProFET, dark blue is the reference. f) KS statistics over ∼ 500 ProFET features, lower is better.

Table 1: Evaluation of ProteoGAN and various baselines with our proposed measures (MMD, MRR and the more powerful gaussian kernel variant of MMD) on the testset. An arrow indicates that lower (↓) or higher (↑) is better. Best results in bold, second best underlined. Given are mean values and standard deviation over five different data splits. Due to the computational effort, the *Unconditional* model was only trained on one split. The positive control is a sample of real sequences and simulates a perfect model, the negative control is a sample that simulates the worst possible model for each metric (constant sequence for MMD, randomized labels for MRR). The second panel shows models without multi-label conditioning, which were conditioned on simplified label sets. Also shown are variants of ProteoGAN that were trained on different datasets with more specific labels.

| Model | MMD↓ | Gaussian MMD↓ | MRR↑ |
|---|---|---|---|
| Positive Control | $0.011 \pm 0.000$ | $0.009 \pm 0.000$ | $0.889 \pm 0.019$ |
| Negative Control | $1.024 \pm 0.000$ | $0.807 \pm 0.000$ | $0.092 \pm 0.003$ |
| ProteoGAN | $\underline{0.040 \pm 0.001}$ | $\underline{0.026 \pm 0.001}$ | $\mathbf{0.545 \pm 0.017}$ |
| Predictor Guided | $\mathbf{0.025 \pm 0.001}$ | $\mathbf{0.017 \pm 0.001}$ | $0.109 \pm 0.005$ |
| Nonhierarchical | $0.228 \pm 0.056$ | $0.155 \pm 0.039$ | $0.271 \pm 0.086$ |
| CVAE (Greener et al.) | $0.166 \pm 0.033$ | $0.108 \pm 0.022$ | $\underline{0.356 \pm 0.027}$ |
| HMM (on label set) | $0.166 \pm 0.003$ | $0.112 \pm 0.002$ | $0.107 \pm 0.006$ |
| n-gram (on label set) | $0.064 \pm 0.004$ | $0.042 \pm 0.002$ | $0.261 \pm 0.029$ |
| Unconditional | $\mathbf{0.032}$ | $\mathbf{0.021}$ | $\mathbf{0.543}$ |
| HMM (on single label) | $0.192 \pm 0.003$ | $0.130 \pm 0.002$ | $0.105 \pm 0.003$ |
| n-gram (on single label) | $0.094 \pm 0.004$ | $0.067 \pm 0.002$ | $0.090 \pm 0.000$ |
| ProteoGAN (100 Labels) | $0.034$ | $0.023$ | $0.470$ |
| ProteoGAN (200 Labels) | $0.161$ | $0.111$ | $0.116$ |
| ProteoGAN (depth-first) | $0.091$ | $0.070$ | $0.270$ |

**Code Availability.** We make source code for ProteoGAN and the evaluations available at `https://github.com/proteogan/proteogan`.

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

# A    APPENDIX

## A.1    BACKGROUND: BIOLOGICAL MECHANISMS UNDERLYING PROTEIN FUNCTIONS

Proteins are complex biological structures but can be simply represented as chains of amino-acids, a 20-character alphabet. While this is useful for modeling approaches, it hides the functional complexity of proteins. In fact, amino acids, with their diverse physico-chemical properties, fold and assemble into complex three dimensional constructs at a local and global level, giving rise to the overall protein structure. In turn, the structure of the protein is responsible for its function, where shape and electrochemical properties dictate the behavior and biological activity of the macromolecule. The link between sequence and function is therefore highly complex and still not understood by the research community. The design of proteins has consequently been so far mainly based on relatively uninformed trial-and-error processes with slight random alterations of the protein sequence and a subsequent assay of functionality (Arnold, 1998) or entail computationally heavy simulations of molecular dynamics (Samish, 2017). However, advances in machine learning have enabled the development of novel *in silico* design methods.

## A.2    EXPERIMENTAL SETUP

In this section, we describe in detail the dataset and preprocessing steps, the baselines and the hyperparameter searches performed.

### A.2.1    DATA

**Sequence data** is acquired from the UniProt Knowledgebase (UniProtKB, Consortium (2019)). The database contains more than 180 million protein sequences with rich annotations such as structure and function. Nonetheless, most of these entries are only automatically annotated. To ensure high quality data for our model, we filter for sequences that are manually curated and have experimental evidence of some form. There are also some specialized proteins that have very long sequences, we only keep the sequences whose length is not exceeding 2048 amino acids, which covers ca. 98.5% of the data points. The resulting dataset contains $149, 390$ sequences. The cut-off at 2048 amino-acids is multiple times longer than other approaches in this field, which is between 30- to 500-long (Das et al., 2018; Davidsen et al., 2019; Greener et al., 2018; Repecka et al., 2019), and allows for a more complete model of the known sequence space.

**Functional labels** are collected from the same database. The gene ontology (GO) resource is composed of three branches, molecular function, cellular component and biological process. We focus on the molecular function ontology, which contains thousands of terms ranging from description like *binding* (GO:0005488) to very specific terms such as *microtubule-severing ATPase activity* (GO:0008568). Each protein is annotated with a set of GO labels describing the molecular function of a protein in modular way. The ontology is structured as a directed acyclic graph with a single root. Further, labels have a hierarchical relationship, i.e. protein with a given functional label inherits automatically the labels of its parents in the DAG (*is-a* relationship). The molecular function ontology resource currently contains more than ten thousand labels, many of which have a highly specific meaning and only few representatives. We therefore restrict the number of labels to the 50 largest classes, the smallest class containing 8659 proteins. We argue fifty labels is sufficient for a proof-of-principle and would even enable the design of experimental assays for validation. Figure A1 illustrates the selected subset of labels and their relationships.

**Train, validation and test splits** were created to preferably represent all labels uniformly in the test and evaluation sets. We use a 80-10-10 split for evaluation in the main body, and a 94-2-2 split for hyperparameter optimization and the results detailed in the supplement. For the validation and test set, we randomly sample sequences until there is at least 1.300 (300 in the optimization) sequences per class. The selections of hyperparameters by the BOHB hyperparameter optimizations for ProteoGAN and by the hyperparameter searches for the baselines are done on the validation set, while the results presented in the main text were acquired on the test set. For sequence sample generation, the model was conditioned on the label combinations of the evaluation/test set and the resulting sequences then compared with the respective set by MMD and MRR.

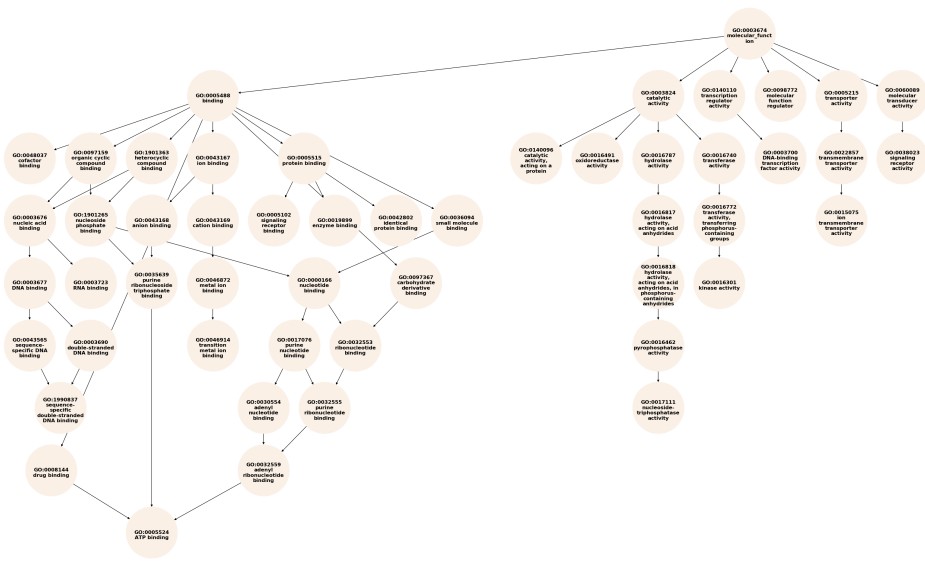

Figure A1: GO DAG of the 50 labels selected for this project.

### A.2.2 BASELINES

We implement four baselines to put the performance of our model into perspective. In this section, we would like to give additional details concerning the two baselines that we gathered from the literature.

PepCVAE (Das et al., 2018) uses a VAE framework with a single layer gated recurrent unit (GRU) RNN for both the encoder and the decoder, in a semi-supervised setting. Conditioning is performed by concatenating the encoded target labels to the latent code. In the paper, the sequences of interest are antimicrobial peptides, with a maximum length of thirty amino-acids. The conditioned on label was binary, i.e. antimicrobial activity or not. The RNN component of the model is highly resource-consuming, therefore we had to trim protein sequences to the first 32 amino-acids to run the model on our data. For a fair assessment, we also run our model ProteoGAN on sequences of 32 amino-acids. We modify PepCVAE by introducing a multiplying factor in front the the KL term of the ELBO loss as suggested by Higgins et al., to increase the stability of the model. We optimise the model with a Bayesian Optimization hyperparameter search, for which we tried $1,000$ combinations of hyperparameters. We do not use BOHB because the early stopping would interfere with the different training phases of the model. The hyperparameters and their value ranges, as well as the final model configuration can be found in Table A1. We refer the reader to Das et al. (2018), Hu et al. (2017) and Bowman et al. (2016) for more information on the model.

CVAE (Greener et al., 2018) uses a conditional VAE (CVAE) in order to generate either metallo-proteins with desired metallo-binding sites or fold properties. In the case of fold properties, the authors introduce iterative sampling and guidance steps in the latent space. The decoder and encoder are both MLPs and the number of layers is chosen with hyperparameter search. Here also, we introduced a KL-balancing term to stabilize training. As for PepCVAE, the model presents a loss scheduling scheme and therefore we could not use the BOHB optimization. However, we performed a Bayesian Optimization hyperparameter search, for which we tried $1,000$ combinations of hyperparameters. Notably, we allowed for an optimization of network architecture by optimizing over the layer numbers for both encoder and decoder, and by optimizing the number of units in the first layer of the encoder and the last layer of the decoder. The unit number then halved towards the latent space with each layer. The hyperparameters and their value ranges, as well as the final model configuration can be found in Table A2. We refer the reader to Greener et al. (2018) for more information on the model.

Table A1: PepCVAE hyperparameters subject to BO optimization.

| Name | | Values | Final Value |
|---|---|---|---|
| Learning rate | | [1e-5,1e-2] | 4e-3 |
| Pretrain iterations | | [1,5000] | 3181 |
| Latent dimension | | $[10,1000]^\dagger$ | 101 |
| Word dropout keep rate | | [0,1] | 0.43 |
| Classifier loss balancing | $\lambda_C$ | $[1e-3,100]^\dagger$ | 9.9e-3 |
| Latent loss balancing | $\lambda_Z$ | $[1e-3,100]^\dagger$ | 1.9e-1 |
| KL balancing | $\beta$ | $[1e-3,100]^\dagger$ | 1.5e-2 |

$^\dagger$ Values were sampled on a logarithmic scale.

Table A2: CVAE of Greener et al. hyperparameters subject to BO optimization.

| Name | | Values | Final Value |
|---|---|---|---|
| Learning rate | | [1e-5,1e-2] | 7.8e-4 |
| Pretrain start | | [1,5000] | 2598 |
| Pretrain end | | [1,5000] | 1251 |
| Latent dimension | | $[10,1000]^\dagger$ | 761 |
| KL balancing | $\beta$ | $[1e-3,100]^\dagger$ | 1.1e-3 |
| Encoder layer number | | [1,5] | 3 |
| Decoder layer number | | [1,5] | 1 |
| Log2(Encoder first layer units) | | [4,10] | 7 |
| Log2(Decoder last layer units) | | [4,10] | 9 |

$^\dagger$ Values were sampled on a logarithmic scale.

The HMM baselines were implemented based on HMMER. For *HMM (on label set)*, all sequences in the training dataset containing a specific label combination were aggregated, for each label set of the test set. For *HMM (on single label)*, all sequences in the training dataset containing a specific label were aggregated, for each of the 50 labels. The resulting sequence sets were aligned with MAFFT (with parameters `--retree 1 --maxiterate 0 --ep 0.123`). Because of the time-intense multiple sequence alignment the sequences sets were randomly sampled to have a maximum size of 5000 sequences. From the alignment, a profileHMM was built with HMMER which was then sampled to generate a sequence.

In the n-gram baseline also, sequences were selected according to label sets and single labels. Here the full data was used. $n$ was set to 3. The sequence lengths were sampled from the training data length distribution.

The *Predictor Guided* baseline was a variant of ProteoGAN without conditioning mechanism trained on the whole data. A sample was generated like in the other models, however the labels were annotated by sampling the per-label probabilities outputted by the NetGO protein function predictor.

Table A3: Evaluation of ProteoGAN and PepCVAE on truncated sequences with length 32. Shown are mean and standard deviation of five different data splits. The arrows indicate that lower ($\downarrow$) or higher ($\uparrow$) is better.

| Model | MMD$\downarrow$ | Gaussian MMD$\downarrow$ | MRR$\uparrow$ |
|---|---|---|---|
| PepCVAE (L=32) | $0.122 \pm 0.019$ | $0.077 \pm 0.012$ | $0.139 \pm 0.022$ |
| ProteoGAN (L=32) | $\mathbf{0.033 \pm 0.002}$ | $\mathbf{0.022 \pm 0.001}$ | $\mathbf{0.321 \pm 0.029}$ |

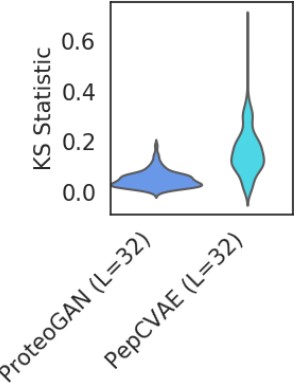

Figure A2: Sequence feature analysis of the models trained on truncated data (Sequence length = 32). KS statistics over $\sim 500$ ProFET features, lower is better.

### A.2.3 HYPERPARAMETER SEARCH

**Description of the BOHB:** For ProteoGAN, we conducted hyperparameter searches with the Bayesian Optimization and HyperBand (BOHB) algorithm. The Hyperband (Li et al., 2017b) algorithm uses successive halving (Jamieson & Talwalkar, 2016) to evaluate a number of models on a given budget of resources. The better half of the models are then evaluated on twice the budget, et cetera. Hyperband is an independent optimization algorithm that has been combined with Bayesian optimization to form Bayesian optimization and Hyperband (BOHB) (Falkner et al., 2018), the optimization strategy used in this project.

**Hyperparameter optimization with BOHB:** We conducted two BOHB optimizations. For both we evaluated $1,000$ models. All networks were trained with the Adam optimizer (Kingma & Ba, 2015) with $\beta_1 = 0$ and $\beta_2 = 0.9$ (following (Gulrajani et al., 2017)). The optimization consisted first of a broad search among 23 hyperparameters and second, of a smaller and more specific search, among 9 selected hyperparameters. For the first BOHB optimization, an optimization iteration was defined as two epochs which we found through pilot experiments was the minimum time to observe a viable trend in the metrics. The parameters $R$ and $\eta$ (in the notation (Li et al., 2017b)) were set to 9 and 3, respectively, which allowed for a maximum training time of 18 epochs (22.5K gradient updates). The optimization objective was to maximize, in the validation set, the ratio of metrics MMD/MRR which are introduced in the main document Section 3.2.1. During the optimization, BOHB selected the models based on evaluations at the end of a training period. For the second optimization, we reduced the number of hyperparameters to only 9. We selected values for the other hyperparameters based on the analysis of the hyperparameter importance of the first optimization (see paragraph below). The hyperparameters that showed either no importance or that were detrimental to training were removed. For this second optimization, the smaller network size allowed for 3 epochs per iteration, resulting in a maximum training time of 27 epochs (1.2K gradient updates). The list of hyperparameters of the two BOHB optimizations and their ranges is presented in Table A4. The parameters of the best models selected by the two BOHB optimizations are presented Table A5.

**Quantification of hyperparameter importance:** After the optimization, we analyze hyperparameter importance with the approach presented in (Hutter et al., 2014). A surrogate model (random forest) is trained on the parameter configurations and the respective evaluation scores. This enables a functional analysis of variance (fANOVA) which allows for a quantification of hyperparameter importance in terms of the variance they explain. It also provides marginal predictions for each hyperparameter which gives insights about their optimal value setting. For the random forest, we used $1,000$ trees with a maximum depth of $64$, and repeat the estimation 100 times. We do so for all evaluated models of the first and second BOHB optimizations. The hyperparameter importances obtained from the first optimization (and resp. second optimization) are presented in Figure A5 (resp. Figure 2). The first fANOVA showed that parameters related to the discriminator (learning

Table A4: Hyperparameters subject to BOHB optimization.

| Name | Symbol | Values |
|---|---|---|
| Use chemophysical properties | | Yes, No, only |
| Label embedding | | one-hot, node2vec, Poincaré |
| Conditioning mechanism | | projection, AC, both |
| AC weighting factor | $\gamma$ | $[1, 1000]^{\dagger}$ |
| Label smoothing factor | $\theta$ | $[0, 0.5]$ |
| Latent noise dimension | $d_Z$ | $[1, 1000]^{\dagger}$ |
| Input noise standard deviation | $\sigma$ | $[0, 1]$ |
| Generator learning rate | $\eta_G$ | [1e-5, 1e-2] |
| Generator learning rate 2 | $\eta_{G2}$ | [1e-5, 1e-2] |
| Discriminator learning rate | $\eta_D$ | [1e-5, 1e-2] |
| Discriminator learning rate 2 | $\eta_{D2}$ | [1e-5, 1e-2] |
| Training ratio | $n_{critic}$ | $[1, 10]$ |
| Learning rate schedule | | constant, cosine, exponential |
| Schedule interval (in epochs) | $i$ | $[1, 18]$ |
| Generator layer number | $n_G$ | $[1, 10]$ |
| Discriminator layer number | $n_D$ | $[1, 10]$ |
| Strides | $s$ | 1, 2, 4, 8 |
| Filter size | $f$ | $[3, 12]$ |
| Generator skip | $h_G$ | $[0, 10]$ |
| Discriminator skip | $h_D$ | $[0, 10]$ |
| Number of projections | $n_P$ | $[1, 5]$ |
| Output source layers | $o_S$ | $[1, 3]$ |
| Output label layers | $o_L$ | $[1, 3]$ |

$^{\dagger}$ Values were sampled on a logarithmic scale. AC = auxiliary classifier.

Table A5: Hyperparameters found in the first and second BOHB optimization. Values with an asterisk indicate the preset configurations in the second optimization.

| Name | First | Second |
|---|---|---|
| Use chemophysical properties | Yes | No |
| Label embedding | one-hot | one-hot |
| Conditioning mechanism | both | both |
| AC weighting factor | 178 | 135 |
| Label smoothing factor | 0.28 | -* |
| Latent noise dimension | 91 | 100* |
| Input noise standard deviation | 0.29 | -* |
| Generator learning rate | 2.0e-3 | 4.1e-4 |
| Generator learning rate 2 | - | -* |
| Discriminator learning rate | 8.5e-4 | 4.0e-4 |
| Discriminator learning rate 2 | - | -* |
| Training ratio | 1 | 1* |
| Learning rate schedule | constant | constant* |
| Schedule interval (in epochs) | - | -* |
| Generator layer number | 2 | 2* |
| Discriminator layer number | 3 | 2* |
| Strides | 4 | 8 |
| Filter size | 8 | 12 |
| Generator skip | - | -* |
| Discriminator skip | - | -* |
| Number of projections | 1 | 2 |
| Output source layers | - | 1* |
| Output label layers | 2 | 1* |

AC = auxiliary classifier.

rate, number of layers) are most important for model performance,[1] and helped to select potentially important hyperparameters for the second analysis. Noticeably, the best model of the first optimization was already a well-performing model but we chose to run a second optimization to better understand the role of key hyperparameters, to gain insight in potential good practice when designing conditional generative adversarial networks and to further improve the performance of our model. The second fANOVA clarified the importance of the remaining hyperparameters, such as use of chemophysical features and label embeddings among others (Figure 2).

We also show marginal predictions for hyperparameters of the first optimization in Figure A6, and for the second optimization in Figure A3 and Figure A4.

**Obtainment of the final model:** The 27 best selected models of the second hyperparameter search were then trained for a prolonged duration of 100 epochs, where the conditioning mechanism and an associated weighing factor became most important, according to the last fANONA study (Figure A4). We evaluated twice per epoch and selected the weights of the final model at the checkpoint that showed the best (smallest) ratio MMD/MRR in the validation set. The final model, ProteoGAN, is a convolutional conditional generative adversarial network, with two conditioning mechanisms: an auxiliary classifier and projections. The dimensions of the convolutional layers are following the pyramidal architecture of DCGAN (Radford et al., 2015), i.e. with increasing output length and decreasing filter depth for the generator, and vice versa for the discriminator. The other hyperparameters are presented Table A5.

---

[1]Some other important factors were learning rate schedule-related parameters such as *Generator learning rate 2* or *schedule*. We realized that these were detrimental to model performance as the short duration of training in the optimization did not allow to estimate long term effects seen in the selected models that were trained for 100 epochs.

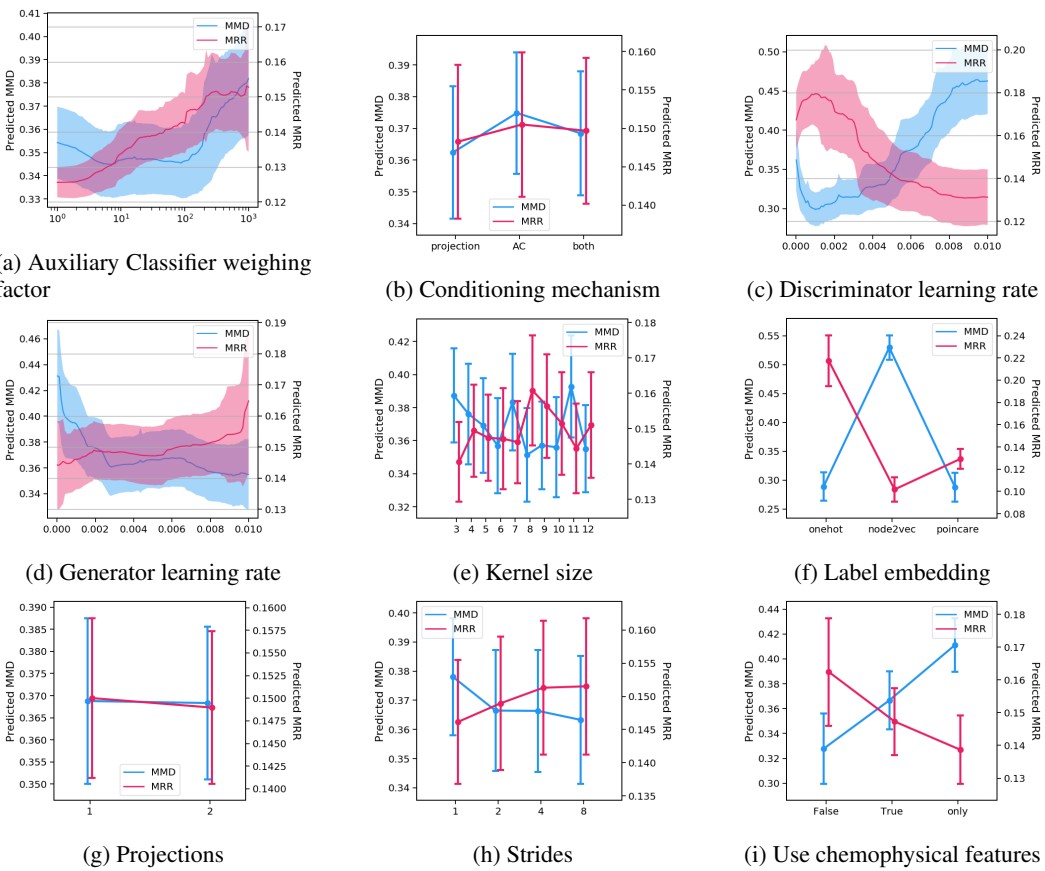

Figure A3: Marginal predictions of hyperparameters based on optimization data in the second optimization. Predictions were obtained training on MMD and MRR. Note that for MMD, lower is better.

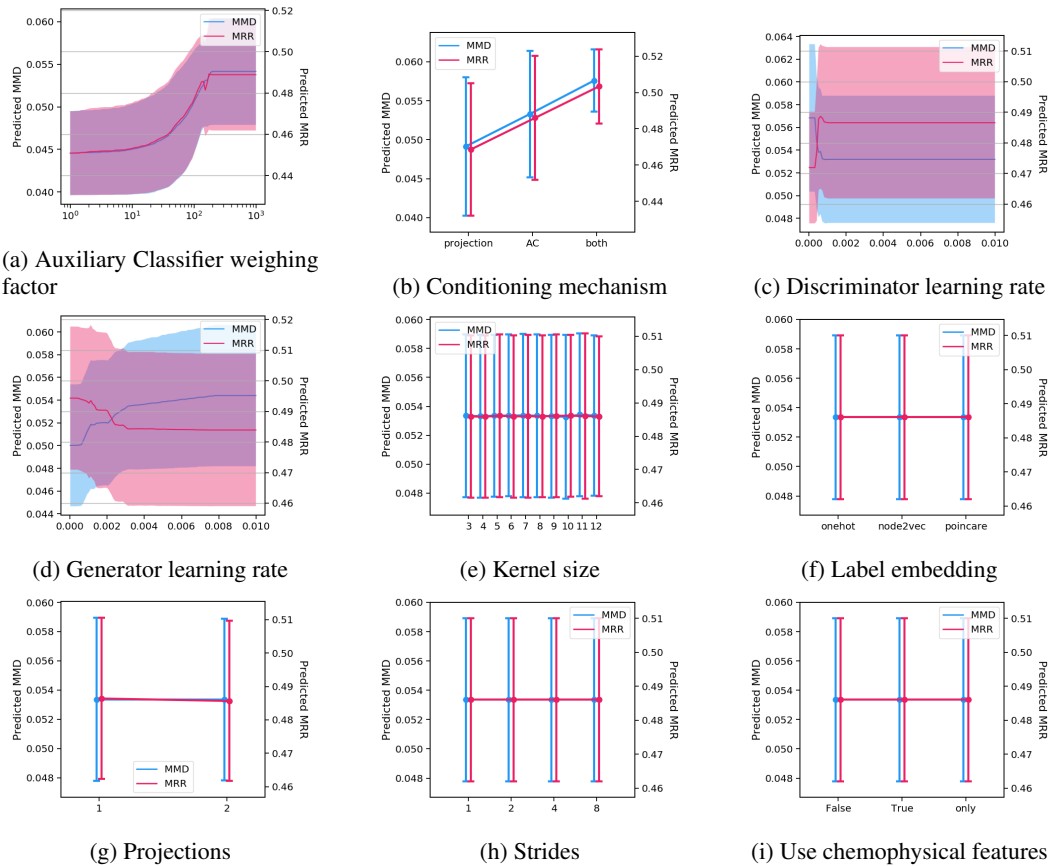

Figure A4: Marginal predictions of hyperparameters based on the data of the 27 best selected models in the second optimization. Predictions were obtained training on MMD and MRR. Note that for MMD, lower is better.

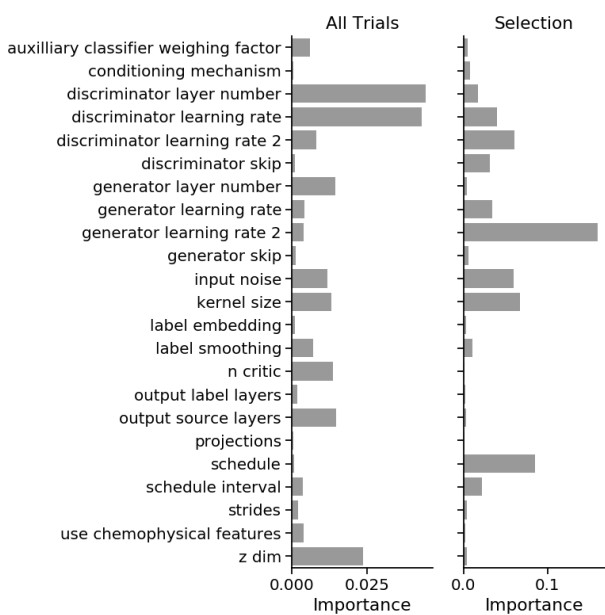

Figure A5: Hyperparameter importance for the first BOHB optimization. Shown are all hyperparameters subject to optimization for all models (left), and a manual selection of models that was trained for 100 epochs (right).

## A.3 METHODS

In this section, we first describe in details the state-of-the-art conditioning mechanisms and constructed variants that we used in this project. Second, we introduce three variants of MRR suited for hierarchically-structured labels. Finally, we assess the proposed evaluation measures of cGANs by estimating empirical worst and best bounds for our experimental setting.

### A.3.1 CONDITIONING MECHANISM

In this section, we first detail how we adapted the Wasserstein loss to the conditional setting, then we describe state-of-the-art conditional GANs' objective functions and variants used in this project.

**Loss function of conditional GANs:** Our models are trained with the Wasserstein objective with gradient penalty from (Gulrajani et al., 2017). As a reminder, the WGAN-GP losses can be written as follows:

$$\begin{aligned}
\mathcal{L}_D &= \mathbb{E}_{q(\boldsymbol{x})}[D(\boldsymbol{x})] - \mathbb{E}_{p(\boldsymbol{x})}[D(\boldsymbol{x})] \\
&\quad + \lambda \mathbb{E}_{m(\hat{\boldsymbol{x}})}[(\|\nabla_{\hat{\boldsymbol{x}}} D(\hat{\boldsymbol{x}})\|_2 - 1)^2] \\
\mathcal{L}_G &= -\mathbb{E}_{q(\boldsymbol{x})}[D(\boldsymbol{x})]
\end{aligned} \tag{3}$$

where $\boldsymbol{x} \sim p(\boldsymbol{x})$ is the data distribution and $\boldsymbol{x} \sim q(\boldsymbol{x})$ is the generator model distribution, $\hat{\boldsymbol{x}}$ is an interpolated sample between a real sequence and a generated one, $m$ is the distribution of interpolated samples, $D$ is the discriminator (or critic), $\mathcal{L}_D$ the loss of the discriminator and $\mathcal{L}_G$ the loss of the generator. The term $\mathbb{E}_{m(\hat{\boldsymbol{x}})}[(\|\nabla_{\hat{\boldsymbol{x}}} D(\hat{\boldsymbol{x}})\|_2 - 1)^2]$ ensures that the discriminator is Lipschitz continuous.

To be able to use the Wasserstein objective with gradient penalty in the conditional setting of projection cGAN (Miyato & Koyama, 2018) (see below), we had to adapt the objective formula to include the label information. Let $(\boldsymbol{x}, \boldsymbol{y}) \sim p$ be a sample from the dataset, where $\boldsymbol{x}$ is the sequence and $\boldsymbol{y}$ the label. Let $D$ be the discriminator and $G$ the generator. Let $q$ be the generator model distribution, such that $\boldsymbol{y} \to q(\boldsymbol{y})$ is defined by the user and $\boldsymbol{x} \to q(\boldsymbol{x}|\boldsymbol{y})$ is learned. In practice, $q(\boldsymbol{y})$ follows the label distribution of the data $p(\boldsymbol{y})$. Let $\hat{\boldsymbol{x}}$ be an interpolated sequence between a real sequence and a generated one. We call $\hat{\boldsymbol{x}} \to m(\hat{\boldsymbol{x}}|\boldsymbol{y})$ the distribution of interpolated sequences given a label $\boldsymbol{y}$. Let $\lambda$ be a weighing factor introduced in (Gulrajani et al., 2017). Taking conditional information into

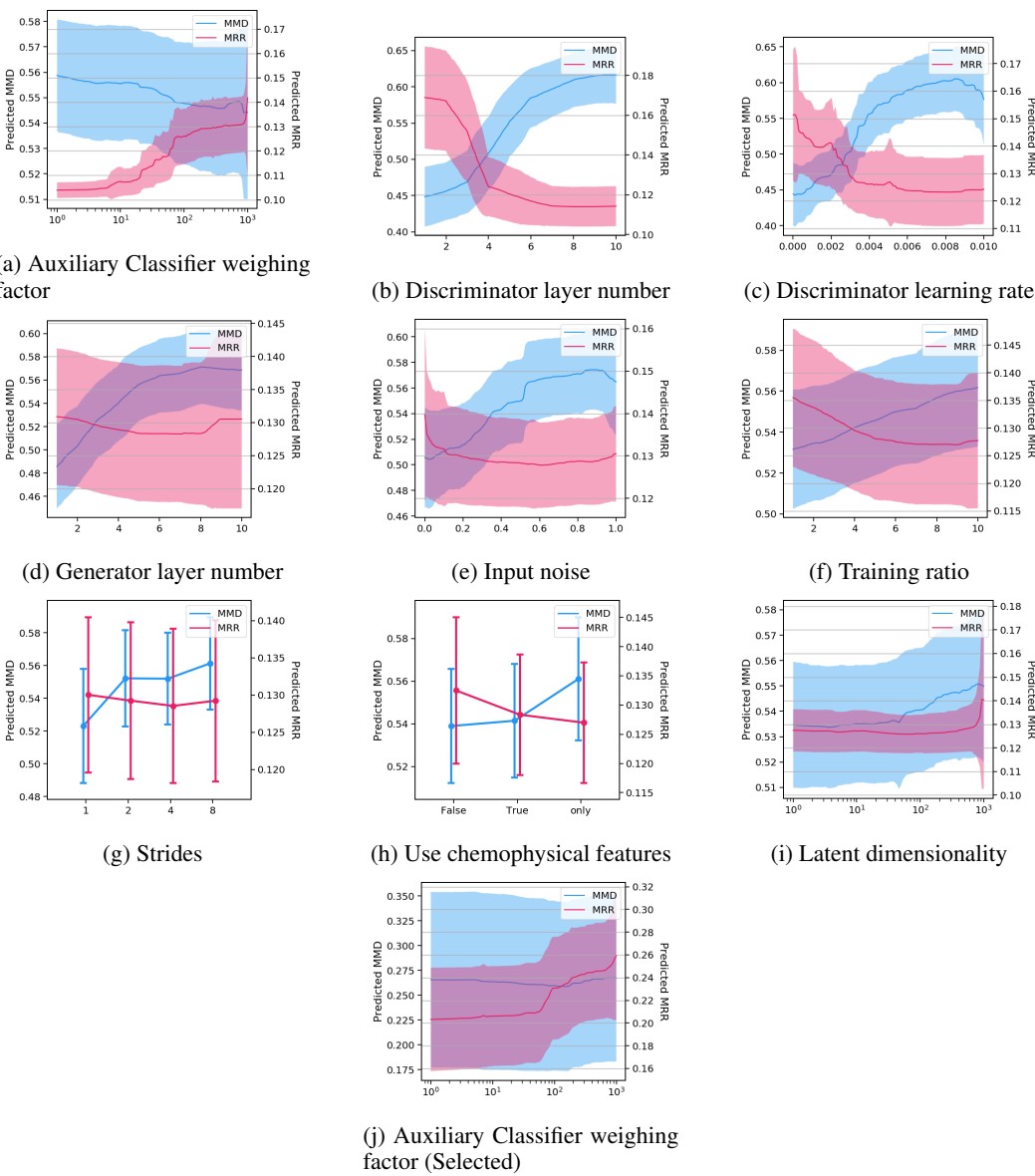

Figure A6: Marginal predictions of hyperparameters based on data in the first optimization. We show some selected predictions that allowed for interpretation, all others were inconclusive. If not otherwise noted, data comes from all trials in the optimization. Predictions were obtained training on MMD and MRR. Note that for MMD, lower is better.

account, the discriminator and generator losses can be expressed as follows:

$$\mathcal{L}_D = \mathbb{E}_{q(\boldsymbol{y})}[\mathbb{E}_{q(\boldsymbol{x}|\boldsymbol{y})}[D(\boldsymbol{x},\boldsymbol{y})]] - \mathbb{E}_{p(\boldsymbol{y})}[\mathbb{E}_{p(\boldsymbol{x}|\boldsymbol{y})}[D(\boldsymbol{x},\boldsymbol{y})]]$$
$$+ \lambda\mathbb{E}_{p(\boldsymbol{y})}[\mathbb{E}_{m(\hat{\boldsymbol{x}}|\boldsymbol{y})}[(\|\nabla_{\hat{\boldsymbol{x}}}D(\hat{\boldsymbol{x}},\boldsymbol{y})\|_2 - 1)^2]], \tag{4}$$
$$\mathcal{L}_G = -E_{q(\boldsymbol{y})}[\mathbb{E}_{q(\boldsymbol{x}|\boldsymbol{y})}[D(\boldsymbol{x},\boldsymbol{y})]].$$

This formulation ensures that the Lipschitz constraints imposed on the discriminator in the unconditional WGAN-GP objective holds for each class.

**Case of the projection cGAN model (Miyato & Koyama, 2018):** In the conditional GAN with projection discriminator model, the discriminator is decomposed into a sum of two terms, one being the inner product between a label embedding and an intermediate transformation of the input, and the second term being solely depending on the input $\boldsymbol{x}$. The new expression of the projection discriminator can be derived by assuming that the label is categorical and that both the log-likelihoods of the data and target distribution can be written as log linear models. Let $\boldsymbol{y} \rightarrow \boldsymbol{v}(\boldsymbol{y})$ be a linear projection of the label into a label embedding. Let $\boldsymbol{\phi}_\theta$ be a vector output function applied to the input $\boldsymbol{x}$ and $\psi_\gamma$ a scalar function applied to the vector output function $\boldsymbol{\phi}_\theta(\boldsymbol{x})$. Let $\mathcal{A}$ be an activation function of choice. The projection discriminator in (Miyato & Koyama, 2018) can therefore be written as:

$$D(\boldsymbol{x},\boldsymbol{y}) = \mathcal{A}(f(\boldsymbol{x},\boldsymbol{y}))$$
$$= \mathcal{A}(\boldsymbol{v}(\boldsymbol{y})^T\boldsymbol{\phi}_\theta(\boldsymbol{x}) + \psi_\gamma(\boldsymbol{\phi}_\theta(\boldsymbol{x}))) \tag{5}$$

The label information is therefore introduced via an inner-product. In practice, the discriminator is equipped with a projection layer that takes the inner product between the embedded label and an intermediate output of the discriminator. This formulation leads to a more stable algorithm compared to a simple concatenation of the label with the input, potentially thanks to the introduction of a form of regularization on the discriminator.

In this project, we also tested the possibility to include several projections in the discriminator. In addition to the previous notations of this paragraph, let us assume that we have $k$ projections. Let $\{g_i\}_{i=1}^k$ be $k$ neural networks, which can be decomposed in $n_i$ layers $g_i = l_{n_i}^i \circ l_{n_i-1}^i \circ \cdots l_2^i \circ l_1^i$. Let $\{p_i\}_{i=1}^k$ be the layer number at which the inner product with the output of the projection $\{\boldsymbol{v}_i\}_{i=1}^k$ occurs in each neural network. The expression of the discriminator is given by:

$$D(\boldsymbol{x},\boldsymbol{y}) = \mathcal{A}(f(\boldsymbol{x},\boldsymbol{y}))$$
$$= \mathcal{A}(\sum_{i=1}^k (\boldsymbol{v}_i(\boldsymbol{y})^T l_{p_i}^i \circ \cdots l_1^i(\boldsymbol{x}) + g_i(\boldsymbol{x}))) \tag{6}$$

In practice we share lower layer parameters and allow for up to four projections. Our BOHB hyperparameter searches did not show evidence of the superiority of projection mechanisms for conditioning purposes when they are the unique type of conditional mechanism in the network. However, the projection models were able to generate sequences similar to naturally occurring ones (low MMD).

**Case of the cGAN model with auxiliary classifier (Odena et al., 2017):** As opposed to projection cGANs, cGANs with auxiliary classifier add a term to the generator and discriminator losses to incorporate the log-likelihood of the correct labels (compare Equation 4). In addition to notations introduced for Equation 4, let $C_D$ be the auxiliary classifier, $ce$ the cross entropy and $\gamma$ a weighting factor. The loss function of cGANs with auxiliary classifiers can be written as:

$$\mathcal{L}_D = \mathbb{E}_{q(\boldsymbol{x})}[D(\boldsymbol{x})] - \mathbb{E}_{p(\boldsymbol{x})}[D(\boldsymbol{x})]$$
$$+ \lambda\mathbb{E}_{m(\hat{\boldsymbol{x}})}[(\|\nabla_{\hat{\boldsymbol{x}}}D(\hat{\boldsymbol{x}})\|_2 - 1)^2] + \gamma\mathbb{E}_{p(\boldsymbol{y})}[\mathbb{E}_{p(\boldsymbol{x}|\boldsymbol{y})}[ce(C_D(\boldsymbol{x}),\boldsymbol{y})]] \tag{7}$$
$$\mathcal{L}_G = -\mathbb{E}_{q(\boldsymbol{x})}[D(\boldsymbol{x})] + \gamma\mathbb{E}_{p(\boldsymbol{y})}[\mathbb{E}_{q(\boldsymbol{x}|\boldsymbol{y})}[ce(C_D(\boldsymbol{x}),\boldsymbol{y})]].$$

$C_D$ typically shares weights with $D$ and is trained when minimising $\mathcal{L}_D$ but is fixed when minimising $\mathcal{L}_G$.

In our work, we compare both types of conditional GANs (GAN equiped with auxiliary classifier or with multiple projections at several layers (see Equation 6)) to a third proposed model that combines both mechanisms. It is important to note that in this case the label information introduced in the projection may not be shared with the auxiliary classifier (compare Figure 1). The fANOVA analysis performed on the second BOHB optimization results shows that the combination of both mechanisms helps to obtain a better performing conditioning mechanism, as measured by MRR.

### A.3.2 EVALUATION MEASURES FOR CONDITIONAL GENERATIVE MODELS IN THE CASE OF HIERARCHICAL LABELS

In addition to the evaluation measures MMD and MRR described in the main document Section 3.2.1, we built three variants of MRR in order to better characterise the effectiveness of the conditional generation and its ability to handle hierarchical multi-label settings. We look at sub-measures of MRR where either all parent nodes ($MRR_P$), all child nodes ($MRR_C$), or both ($MRR_B$) are ignored in the ranking of a conditional MMD term. Removing parent or child terms attempts at understanding to which degree the conditioning mechanism leads to severe off-target generation of sequences of unrelated labels. Indeed, these three alternative measures do not penalize if the generated distribution of sequences for a given label is closer to either parents' or children's sequences than it is to its target's sequences, which is less severe than off-target generation of sequences of an unrelated label. Therefore, getting good $MRR_X$ values would indicate that our model is able to conditionally generate sequences up to closely related (parent's or child's) functions. Additionally, comparing $MRR_X$ values between themselves and with MRR would give some insight in closely-related conditional generation performance. For example, a $MRR_P$ (resp. $MRR_C$) value much larger than MRR could indicate that sequences that are generated with a target function are often closer to the natural sequences exhibiting the parent (resp. child) functional label, i.e. the model is too general, or too specific, respectively.

### A.3.3 ASSESSMENT OF THE EVALUATION MEASURES FOR CGANS

We assess the quality of the proposed evaluation measures by constructing "best case" and "worst case" scenarios, to understand what would represent a perfect success or failure mode of our model. We consider as "best case" the case where the generative model generates sequences that are observed. The "worst case" scenario differs depending on the evaluation measure.

**Scenarios for MMD:** MMD has theoretical bounds of $[0, \sqrt{2}]$ if the sequences in both sets are self-similar and totally dissimilar from each other. As we aim to compare real sequences to generated sequences that resemble real sequences, we therefore fix one set to be a collection of $n$ natural protein sequences and the second set to be $n$ other natural protein sequences modified with different percentages of random noise. In practice, the set of natural sequences is the test set used to report the results in the main document, and the random noise is injected in the form of single-point mutations to sequences of the second set. The results are reported in Table A6 and indicate that MMD is a proxy for the quality of the generated sequences. We observe that MMD increases with the amount of noise injected in the sequences of the second set. The generation of close to constant sequences is a plausible failure mode of the GAN and would lead to a very high MMD value (last row). The lengths of the sequences of the mutated set were conserved, however we also report MMD with respect to fully random sequences of maximum length. The MMD value between two sets of real sequences is around 0.0237, adding 1% of noise to the sequences in one of the set leads to an MMD value of 0.0240, 10% of noise to 0.0324 and 20% of noise to 0.0484. In comparison, in biology, proteins have been shown to be viable up to 30 - 60% of mutations in the amino-acids of their sequences (Repecka et al., 2019; Markiewicz et al., 1994; Ng & Henikoff, 2001). We also report empirical p-values following Borgwardt et al. (2006), under the null hypothesis that the two sets are from the same distribution. These were obtained by ranking the original MMD statistic in 1000 iterations of statistics where the aggregated sequences were randomly assigned to each of the two sets.

**Scenarios for MRR:** Since MRR is a conditional measure, we constructed the "worst case" sample as a set of natural protein sequences with randomized label assignments. This aims to simulate a generative model that produces well-formed sequences, but ignores the conditioning objective. One could also construct a scenario that simulates an antagonistic model that actively assigns wrong labels, instead of random ones. This will likely not occur in practice, though. Table A7 shows the MRR values for a real data sample and the same sample with randomized labels. The reference for MRR was again the test set and the evaluated sample an equally structured set ("Positive Control", Table A7) where the label annotations were randomly shuffled among the sequences ("Negative Control", Table A7). The MRR evaluates a set of sequences with respect to the 50 selected labels. We also look at the sub-measures of MRR where either all parent terms ($MRR_P$), all child terms ($MRR_C$), or both ($MRR_B$) are ignored in the ranking of a term. This gives additional insights on how well the model works with respect to the up- and downstream labels in the GO DAG.

| Sample | MMD | p-value |
|---|---|---|
| Dataset Sample | 0.0237 | 0.1499 |
| Dataset Sample + 1% noise | 0.0240 | 0.0370 |
| Dataset Sample + 2% noise | 0.0243 | 0.0050 |
| Dataset Sample + 3% noise | 0.0248 | 0 |
| Dataset Sample + 5% noise | 0.0262 | 0 |
| Dataset Sample + 10% noise | 0.0324 | 0 |
| Dataset Sample + 20% noise | 0.0484 | 0 |
| Dataset Sample + 30% noise | 0.0660 | 0 |
| Dataset Sample + 50% noise | 0.1009 | 0 |
| Dataset Sample + 100% noise | 0.1788 | 0 |
| 100% noise (maximum length) | 0.3044 | 0 |
| Constant (all leucine) | 1.0258 | 0 |

Table A6: MMD values with different percentage of mutations, p-values

Table A7: Best and worst case MRR, as well as model evaluations of the main text with the extended set of MRR measures. *Positive Control* simulates a perfect model, *Negative Control* a model that ignores conditional information.

| Model | MRR | $MRR_P$ | $MRR_C$ | $MRR_B$ |
|---|---|---|---|---|
| Positive Control | 0.7887 | 0.8260 | 0.8520 | 0.8925 |
| Negative Control | 0.0909 | 0.1177 | 0.0923 | 0.1196 |
| ProteoGAN (ours) | $0.5956 \pm 0.0237$ | $0.7018 \pm 0.0180$ | $0.6494 \pm 0.0237$ | $0.7588 \pm 0.0166$ |
| Unconditional | $0.5219 \pm 0.0195$ | $0.6089 \pm 0.0242$ | $0.5729 \pm 0.0205$ | $0.6643 \pm 0.0225$ |
| Predictor-guided | 0.1071 | 0.1367 | 0.1261 | 0.1562 |
| Greener et al. | $0.3132 \pm 0.0161$ | $0.3658 \pm 0.0157$ | $0.3775 \pm 0.0154$ | $0.4306 \pm 0.0150$ |
| PepCVAE (L=32) | $0.1910 \pm 0.0137$ | $0.2038 \pm 0.0156$ | $0.2103 \pm 0.0195$ | $0.2267 \pm 0.0194$ |
| ProteoGAN (L=32) | $0.3159 \pm 0.0205$ | $0.3464 \pm 0.0207$ | $0.3532 \pm 0.0240$ | $0.3892 \pm 0.0221$ |

**Results for MRR variants:** Table A7 shows the results for the MRR variants for some models. The results confirm that our model is better at conditional generation than the baselines, including the baseline that consists of training 50 unconditional models. The comparison between MRR variants suggests that proteins often resemble proteins in the target class. When this is not the case, proteins are often similar to their parent class, which makes sense as the class is then more general. The small difference between the $MRR_B$ value of our model and of the positive control indicates that the model rarely generates sequences that resemble proteins in an unrelated class. Additionally, compared to the controls, the $MRR_C$ are relatively low compared to MRR, which suggests that the model does not tend to create more specific child labels, which would be detrimental in a biological application.

### A.3.4 LOSSES AND REAL-TIME EVALUATION OF THE FINAL MODEL

The loss function of the final model presented in the main document, combining projection and auxiliary classifier, is shown Figure A7. We monitored the duality gap (red), for which we split the training data into an adversary finding set and a test set of 1% of the train set each. The duality gap is well-behaved, with a fast convergence to 0, indicating that there is no mode collapse and suggesting that the samples are of reasonable quality. Also, the evaluations of MMD and MRR can be seen during training (evaluated twice per epoch) which provides valuable information for model selection and early stopping.

### A.4 RESULTS WITH A SMALLER TESTSET SIZE

Table A8: Evaluation of ProteoGAN and various baselines with our proposed measures (MMD and MRR) and NetGO ($F_{max}$) on a smaller testset testset (n=300, ca. 2% of the data). An arrow indicates that lower ($\downarrow$) or higher ($\uparrow$) is better. Given are mean values of $n = 10$ (MMD/MRR) and $n = 3$ ($F_{max}$) different random seeds for the latent variable of the model. Note that models marked with (L=32) have been trained and evaluated on a set of truncated sequences and are hence not directly comparable to the other values. Also, since without multi-label conditioning, the *Unconditional* model was conditioned on different label sets as the other models and controls.

| Model | MMD$\downarrow$ | MRR$\uparrow$ | $F_{max}\uparrow$ |
|---|---|---|---|
| Positive Control | 0.0237 | 0.7887 | 0.7705 |
| Negative Control | 1.0258 | 0.0909 | 0.3485 |
| ProteoGAN (ours) | $0.0463 \pm 0.0003$ | $\mathbf{0.5956} \pm 0.0237$ | $0.4178 \pm 0.0004$ |
| Unconditional | $\mathbf{0.0380} \pm 0.0010$ | $0.5219 \pm 0.0195$ | $0.3050 \pm 0.0024$ |
| Predictor-guided | 0.0428 | 0.1071 | $\mathbf{0.4776}$ |
| Greener et al. | $0.1611 \pm 0.0012$ | $0.3132 \pm 0.0161$ | $0.4658 \pm 0.0020$ |
| PepCVAE (L=32) | $0.1504 \pm 0.0054$ | $0.1910 \pm 0.0138$ | $0.4140 \pm 0.0003$ |
| ProteoGAN (L=32) | $0.0372 \pm 0.0005$ | $0.3160 \pm 0.0205$ | $0.4147 \pm 0.0005$ |

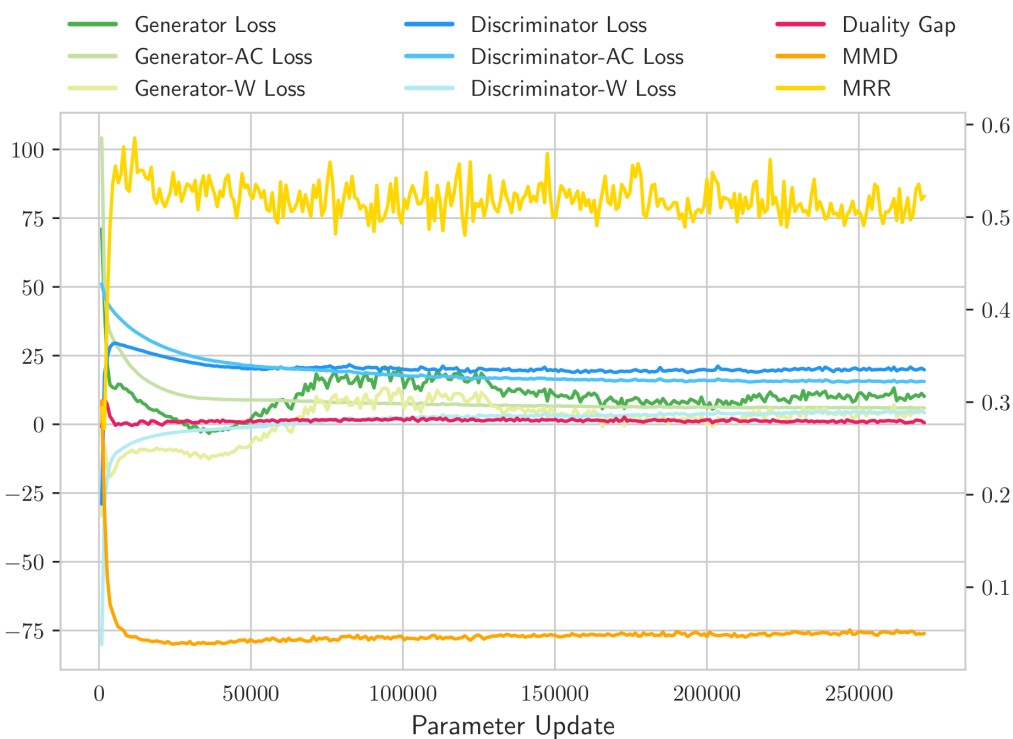

Figure A7: Losses and evaluations at training time. W = Wasserstein, AC = Auxiliary Classifier

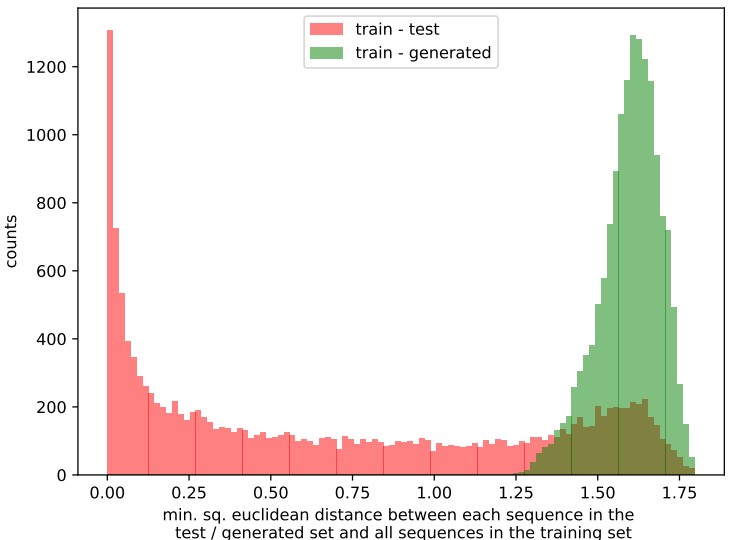

Figure A8: Distributions of pairwise distances in kernel feature space between a testset and the training set (red) and a generated set of ProteoGAN and the training set (green). It can be seen that the generated sequences are not closer to the training set than the testset (which would indicate overfitting). Further the generated sequences are about as far, but not further, away from the training set than the testset.

