# OpenReview forum: "Conditional Generative Modeling for De Novo Hierarchical Multi-Label Functional Protein Design"
_ICLR.cc/2021/Conference — Reject_

### Official Review · AnonReviewer2 · 2020-10-28
**Interesting conditional generative model, but lacking proper validation criteria**

**Rating:** 3
**Confidence:** 4

**Review:**

Brief Summary: The authors describe a method of generating sequences from neural networks conditioned on sequence annotations using novel statistics about the distribution of sequences. However, the evaluation criteria are lacking, requiring additional work.

Pros:
- I think using MMD to determine the similarity between the true and generated distributions was very interesting. I am also a big fan of the kmer kernel for the MMD test statistic.

-I also think understanding mode collapse with the duality gap was a good idea.

- I appreciated the hyperparameter optimization studies to understand which parameters were important to tune to get working.

Cons:
- I am concerned by the repeated usage of "valid protein sequences" throughout the paper. I'm not sure if the authors intended to use the word "functional". The real test of "valid" would be experimental validation in the lab to determine if the sequences still performed the functions of the GO categories, but I understand that this is not a particularly viable option. However, to make such a claim, the authors could calculate various energy statistics from Rosetta, or other mutation effect predictors to determine if the sequences are likely to be functional.
- I am particularly troubled by the statement in the paper: “With respect to general sequence quality, ProteoGAN reaches MMD values of 0.0463, which corresponds to roughly 20% of random mutations in a set of natural sequences (compare supplementary Table A5). In comparison, previous in vitro biological experiments showed that proteins with 34% (resp. 59%) of mutated positions compared to the wild-type were viable…”
This, in no way, proves that sequences generated by this method are valid or functional. A single mutation can abrogate protein function, let alone twenty.
-I am also concerned about the train-test-validation splits in this paper: It seems like this was done randomly. This is not how to prove the method can generalize to new sequences. I worry that generated sequences in the training set can have homologues in the test set. Train/test splits using CATH or SCOP would be more appropriate.
-Since there is no control for homology in this work, it would be interesting to just use HMMER generated sequences of a family in a given GO category to compare against (simplest baseline model).
-In Figure 5, I don't understand where the "Random" values came from. Are you sampling from sequences of the same length or kmer composition? Truly random amino acids is not a useful comparison.
- MRR is an odd statistic. Why not just BLAST, or an alignment-based approach? How does MRR compare to e-value, or closest in sequence ID?

Neutral:
- I am surprised the authors are only using manually annotated sequences for this task, or not leveraging some sort of pretraining. Homologous sequences could improve model performance.
- As someone working in biotechnology, I think there is great interest in the refinement of molecular properties to highly-specific functions, and little interest in generating sequences generally from GO categories, which are much too broad.

---

> ### Author Response · Authors · 2020-11-23
> **Thank you for your feedback, we hope to have addressed your concerns**
>
> We are very grateful for your positive and constructive feedback and hope that our comments below can address some of your concerns.
>
> “[...] repeated usage of "valid protein sequences" [...] determine if the sequences are likely to be functional.”
>
> We acknowledge that there has been a terminology misunderstanding here. We refer to the global answer for a better definition of terms. We confirm that we did not experimentally validate any of the generated sequences and, for experimentally verified sequences, we reserved the term “functional”. We explicit this further in the revised version of the manuscript. The suggestion of assessing the energy assessments of the generated sequences in Rosetta is a good approach to validate sequences before lab experiments but is not suitable for a high-throughput assessment.
>
> “[...] “With respect to general sequence quality, ProteoGAN reaches [...]” A single mutation can abrogate protein function, let alone twenty.”
>
> Since we do not test the proteins we generate in a lab, we can indeed not verify that the proteins are functional for sure. However, we use several measures (MMD, MRR) to check similarity of the generated sequences with real sequences in distribution and conduct bioinformatic experiments (ProFET pipeline) to show that the proteins that are generated, while potentially showing a high mutation rate compared to existing sequences, are plausibly functional/valid because exhibiting a large number of similar characteristics to real sequences. In the current version of the paper, we changed the terminology to describe these sequences as “validated in-silico” (see previous comment and global answer). The objective of this sentence “With respect to … viable” was to state that a high mutation rate is not a necessary condition for a protein to be non-viable, but we are aware that what matters is much more than a simple count of mutations. We rephrase this sentence in the paper to make it clearer.
>
> “ [...] train-test-validation splits in this paper [...].”
>
> We thank the reviewer for the important comment. We clarify this point in the global answer as it was raised by several reviewers.
>
> “[...] use HMMER generated sequences of a family in a given GO category [...].”
>
> We added such a baseline to our results table.
>
> “In Figure 5,[...] Are you sampling from sequences of the same length or kmer composition?”
>
> Here random means that we used the same model architecture as ProteoGAN to generate sequences, but use the random weights used for initialisation without training the model. This was to show the scale on which features should be compared (to answer: “Is this difference I see large or small?”). However, we removed this baseline from the set of baselines as it brought some confusion and was clearly underperforming compared to other models (and compared to the new baselines).
>
> “MRR is an odd statistic. Why not just BLAST, or an alignment-based approach? How does MRR compare to e-value, or closest in sequence ID?”
>
> We used MRR between GO-specific MMDs in order to be able to evaluate the conditioning mechanism of our model in a fast manner. Based on Table 1, the metric enables us to rank models in terms of their ability to generate sequences specific to a given GO label. Additionally, we wanted a distribution-based measure since we are assessing generative model performance, not individual sequences. Further, MRR can be used with a wide range of sequences. Generated sequences are often very dissimilar to natural proteins in primary sequence while being similar in their characteristics. Alignment-based methods will most often fail to find any significant hit. Using alignment-based methods would restrict our target space of sequences to close homologs in nature (which we want to avoid), and such a measure would provide no information in regimes that are a bit further away from known sequences. This however is needed to guide the construction of models that are initally not good enough towards a better parameter space in an optimization (as we did).

---

### Official Review · AnonReviewer1 · 2020-10-28
**Review for "Conditional Generative Modeling for De Novo Hierarchical Multi-Label Functional Protein Design"**

**Rating:** 4
**Confidence:** 4

**Review:**

The paper applied conditional generative adversarial networks (cGANs) on the task of protein sequence design with respect to GO molecular functions.

Overall the contribution was based on empirical numerical experiments. It was good practice yet the contribution or the novelty seemed fairly limited.

Understood that one challenge of evaluating protein sequence design method at a large scale is the high cost of wet lab experimentation. Yet it would be nice to provide case studies, either through small scale lab experiments or deep diving into example outputs of the proposed method.

In addition, the experiment setup specified that the number of labels was restricted to 50 most common CF; as a result, the selected CFs would all be very high level and hence lack of specificity. It’d be good to see how the propose method would generalize to more specific CF terms.

Lastly, GO is a directed graph; yet it’s unclear whether the authors used the 50 CF terms as independent labels or in a DAG fashion. It’d be nice to incorporate the correlation among labels into protein design.

---

> ### Author Response · Authors · 2020-11-23
> **Thank you very much for your comments, we hope to have addressed your points**
>
> Thank you for your insightful comments on our work. We have included some of your suggestions in our revised submission. Here below we address them in order.
>
> "Understood that one challenge of evaluating [...] deep diving into example outputs of the proposed method."
>
> We agree that wet lab experiments would be of interest, but before diving into experiments we needed to ensure some theoretical foundations (i.e. a working conditioning mechanism, stable architecture for GANs, evaluation measures to assess model performance, etc) which we provide with this submission. Future work could be concerned with more biologically focused evaluation of the model outputs.
>
> “In addition, the experiment setup specified that the number of labels was restricted to 50 most common CF [...].”
>
> We have added models trained on the most common 100 and 200 labels, as well as a “depth-first” selection of 49 labels to show how the model would behave with more specific terms. We note, however, that the aim of our work was to focus on more general terms, to lay the foundation for de-novo design of unseen label combinations. There exists plenty of work addressing the refinement of protein sequences towards more specific functions. We have revised our introduction to state the motivation for this project more clearly.
>
> "Lastly, GO is a directed graph [...] incorporate the correlation among labels into protein design."
>
> We use a range of embeddings to incorporate the label structure into our generative approach. Also here we have revised the text to describe data and model structure more clearly. Please also see the comparison to a baseline without this label structure information (nonhierarchical).
>
> We hope that these revisions will increase the value of our work for the broader community and look forward to hearing any additional thoughts you might have on the paper.

---

### Official Review · AnonReviewer4 · 2020-10-29
**This is a well executed protein design project with several good ideas but also several important missing pieces**

**Rating:** 7
**Confidence:** 3

**Review:**

This manuscript describes the development of a conditional GAN (ProteoGAN) for performing protein design.  The key idea is to design sequences while taking into account the hierarchical structure of the Gene Ontology.  Other contributions include the development of novel measures of assessment in this domain.

This is an ambitious piece of work.  Just describing everything that was done took a lot of space, and the authors have done a good job of relegating some of the important but less critical details to the supplement.

Overall, the experimental design seems to be quite rigorous.  For example, it is nice to see significant time spent on hyperparameter optimization for the competing methods.

One of the main contributions of the paper is to propose a couple of performance measures that allow a set of designed proteins to be compared to a set of reference proteins. This is done using the MMD measure, based on a 3-mer spectrum kernel.  While the measures themselves seem reasonable, in order to propose a novel measure it seems that the paper should include some meta-evaluation of the measures themselves.  This could be achieved, e.g., by using 3D structure information to assess how well the proposed measures capture "real" similarity between proteins.  As it stands, the paper simply proposes the measures and then evaluates various protein design methods using those measures.

The general idea of taking into account the hierarchical structure of the GO is sensible, but not much effort seems to go into evaluating how well this worked.  Three embedding modes are mentioned briefly on p. 4, but the embeddings themselves are not discussed in the Results section, and there is no comparison to a variant of the proposed method that does not use this hierarchical information.

The results section also mentions, with respect to CVAE, that "close inspection of the sequences and of the predicted chemophysical properties (Figure 5) of the sequences indicates that the sequences are not well-formed."  I can see from the figure that the sequences exhibit unusual characteristics, but more work should have been done to delve into this.

Minor: In the second sentence of the abstract, I did not initially understand the meaning of the prhase "or sequence fragments."  Based on the recurrence of this phrase on p. 2, I gather that the authors are talking about scenarios in which a protein is given a sequence fragment as the starting point for optimization, but this should be clarified.

---

> ### Author Response · Authors · 2020-11-23
> **Thank you very much for your feedback and the interesting ideas, we hope to have addressed some of your points**
>
> We would like to thank you for your positive feedback and interesting suggestions. We will here try to address some of your concerns:
>
> “While the measures themselves seem reasonable, [...] how well the proposed measures capture "real" similarity between proteins.”
>
> We assess the suggested measure by evaluating it on real sets of sequences and observe that it behaves like we expect it (see Figure A6), and we also observe that MMD correlates well with a large amount of biologically relevant sequence-level features (ProFET, Figure 5),  resulting in a good measure of generative performance at the sequence level. Linking our MMD similarity measure to structurally-derived similarities is a very interesting suggestion and we will further explore it in future work.
>
> “Three embedding modes [...] method that does not use this hierarchical information.”
>
> We did observe that one-hot encoding performs better than the other embedding schemes, we refer to the paragraph “Insights on cGANs architecture” for the comment on the best label embedding. Moreover, we agree that a comparison with a baseline which does not account for hierarchical nature of the label is of interest. We therefore added a comparison between the performance of our model and the one reached by a model trained on one-hot encodings of the GO terms with removal of the parent terms (Nonhierarchical model, in Table 1). We notice that the hierarchical information of the labels enables better performance across our three evaluation criteria and more realistic sequences. We added these observations to our discussion in the paper.
>
> “Minor [...]."
>
> We clarified the content of the abstract to avoid any misunderstanding.

---

### Official Review · AnonReviewer3 · 2020-11-03
**Reasonable idea, but experiments are unconvincing**

**Rating:** 3
**Confidence:** 5

**Review:**

In this manuscript, the authors present a conditional GAN for generating protein sequences given specified GO terms. They argue that this approach to conditional protein generation is more appropriate than sequence-based generation, because it gets directly at functional specification. At a high level, this is an interesting idea though it has already started to be explored by other works. The authors are correct that these works focus primarily on optimize a single function of interest. However, there doesn’t seem to be any specific reason that guided design approaches could not generalize to multiple criteria. Regardless, controlled generation of proteins with pre specified functions is certainly interesting.

That said, the work presented here is too preliminary with too many missing baselines, missing or poorly/confusingly described experiments, and the dataset is not fully described. No comparisons are made against HMMs or other typical autoregressive generative models. Furthermore, there appear to be critical problems with the experiments. Specific comments follow below (in no particular order).

1.	If I understand correctly, the MMD method used by the authors compares k-mer distributions between generated and real sequences. They set the k-mer size to 3 (“The size of the k-mers was set to 3”). Therefore, MMD is only measuring whether the sampled 3-mer frequency matches the observed 3-mer frequency. This metric seems too simple and can’t capture complex dependencies that exist in protein sequences. In fact, a simple 3-gram sequence model can perfectly optimize this metric. A simple baseline model in which sequences are generated from a 3-gram model conditioned on the function labels would be informative here. No evidence is otherwise provided that MMD is a good measure of the quality of a generative model.
2.	The GAN seems like overkill for this problem. Have the authors considered any autoregressive generative models that can be trained by maximum likelihood?
3.	GANs also have the problem that likelihoods cannot be computed which makes it difficult, maybe even impossible, to use them to rank candidates or otherwise as priors over sequence space. If one wanted to use this model in practice, how would I choose a set of sequences to synthesize and experimentally validate? Strictly by random sampling? Is there a way to choose the top-k most likely sequences? In practice, experimental throughput is not large enough to explore many random samples, which is why focused design approaches have received so much attention. Is there a way to resolve this with GAN-based models?
4.	Train/val/test splits: the val and test splits are very small. Much smaller than typically used for evaluating ML models (only 2% for val and test). Also, there is no analysis of sequence complexity within these groups. Typically, models are only of interest if they can generalize to relatively distant sequence. How similar are the train/val/test distributions?
5.	Table 1 reports standard deviations (I assume. It isn’t stated what the error values are in the caption.) for random samples from the latent variables, but this doesn’t take into account variability due to model training or the small test set. I encourage the authors to report standard errors over multiple data splits instead.
6.	I propose a simple generative model: given some set of GO terms, retrieve proteins with those terms from the training set. Then, to generate new proteins I simply propose those sequences. As a second approach, I fit an HMM to those sequences and sample new sequences from the HMM. How does this compare to the proposed approach? These baselines would also reveal how easy the val/test sets are due to similarity to the training set.
7.	In Table 1, “Positive Control” is described confusingly: “The reference for MRR was again the test set and the evaluated sample an equally structured set (”Positive Control”)” on p.23 is the only definition for positive control I can find. What is the positive control?
8.	The primary use case for a model like this seems to be the ability to generate proteins with combinations of functions not found naturally. Otherwise, choosing a protein to start from is trivial: retrieve it from the database. Have the authors considered that use case?
9.	Sequences lengths for the baseline models were limited to 32 amino acids whereas the proposed model has a maximum sequence length of 2048. Not only that, but only the first 32 amino acids were used? This does not seem like a fair comparison at all. Typical proteins are much longer than 32 amino acids, so it isn’t surprising that the baselines perform poorly. Furthermore, this limitation makes no sense. RNNs can easily be applied to protein sequences of length 2048.
10.	Proteins can have multiple GO terms associated with them. How was that dealt with? Do these proteins occur multiple times in the dataset for each term? Are proteins generated conditioned only a single term or multiple terms simultaneously?
11.	As far as controlled sequence generation is concerned, “Generative Models for Graph-Based Protein Design” - Ingraham 2019 seems highly relevant but is not mentioned.

Things that would improve my rating:
1.	Fully describe the train/val/test splits with analysis of sequence similarity between splits within functional categories. Increasing the size of the val and test splits is also a good idea.
2.	Add reasonable baselines to the model evaluation and provide proof that MMD with k-mer size of 3 is a good measure of generative model quality. Baselines must be trained to generate complete sequences, rather than only the N-terminal 32 amino acids.
3.	Provide a more rigorous description of the model and experiments. A reader should, ideally, be able to understand the conditioning structure from the main text.

---

> ### Author Response · Authors · 2020-11-23
> **Thank you for your review and your suggestions, we hope to have addressed your concerns**
>
> We warmly thank you for the in-depth review and insightful feedback on our work. We have added several baselines and revised the textwork to address your points. Here below and throughout our revised submission, we answer your comments.
>
> “However, there doesn’t seem to be any specific reason that guided design approaches could not generalize to multiple criteria.”
> To the extent of our knowledge, there exists no guided design method that would freely generate proteins with different types of functions. Guided design approaches usually depend on predictors, which are often only available for specialised functions but not general purpose. Additionally, we believe that cGANs are essentially enabling the generalisation of guided design to multiple criteria with a predictor trained at the same time.
>
> 1. Indeed MMD measures the discrepancy between embeddings, in this case from a 3-mer string kernel. While the practical application seems very simple, we note that there is rigorous theory supporting MMD as a test statistic (Gretton et al). It was first proposed for two-sample tests of biological data (Borgwardt et al.) and this is exactly what we need for assessing the performance of a GAN whose objective it is to model the distribution of proteins. To assess performance, we are only presented with two sets of samples, one generated and one real (the dataset). It follows naturally to perform a two-sample goodness-of-fit test. Luckily, the kernel framework enables this on structured data (the protein sequence strings). We use the kernel proposed by Leslie et al., which, they demonstrated, performed comparable to classic methods such as PSI-BLAST on remote-homology detection. We added that all functional classes we are concerned with can be linearly separated in the 3-mer embedding space (94% accuracy on all annotations). (Further note, that even with k=3, the dimensionality of the embedding space is already 8000.) This should show that the 3-mer kernel is a sufficient choice for our setting, for proofs of MMD we refer you to Gretton et al. Of course there are more powerful kernels, but we stress that we chose this particular kernel because it can be efficiently computed and hence be used to monitor training and to guide large optimizations. Unfortunately, whether or not the MMD statistic can discriminate between functional and non-functional sequences cannot be determined as the required effect size (or a H1 distribution) is not available. Our phrasing of “valid sequences” was therefore slightly premature, we have changed the text to make clear what we mean by that. We have also added an n-gram baseline, and revised text parts concerning MMD.
> 2. We have considered multiple generative frameworks and model architectures. As you have pointed out, proteins are very complex and hence we chose a powerful model. Further, we require multi-label conditioning, which is directly possible within the framework of GANs. We have added several baselines based on HMMs and n-gram models to show that this choice was justified.
> 3. We are not aware of a method that would enable likelihood estimation in GANs, at least not when the generator is based on a convolutional architecture (for an RNN-based generator, one could use the likelihood of the generated sequence). However, the auxiliary classifier of our model enables a ranking of sequences, thanks to the classification scores it yields for the function of interest.
> 4. We have increased the size of the data splits to a more standard 80-10-10 which increases the power of the MMD estimate. We provide analysis of feature similarities between sets in supplemental Figure A8.
> 5. Yes, these are standard deviations. Thanks for pointing this out. We also agree that several data splits are a better assessment, we have retrained all models on 5 different splits.
> 6. Concerning the second approach, we added HMM baselines as you proposed, thank you for your suggestion. As you can see, the sequences generated by our model outperform the others across all our criteria. For the database retrieval baseline point, please see our answer to 7 (where “positive control” corresponds in fact to retrieving sequences from the training set) and 8 below.
> 7. The positive control is a set of natural sequences from the training set with a label distribution that is similar to the testset label distribution. One could see it as a second test set, which aims to simulate a perfect generative model (that would generate real proteins). We agree that this was not clear enough in our initial submission and have revised the text to better explain these two baselines.
> 8. Indeed, eventually this is the goal of our work. Simply retrieving sequences from the database would not allow for novel designs. For more on this, please see our general answer to all reviewers as this was a common confusion we hope to address with the current revision.

---

> > ### Author Response · Authors · 2020-11-23
> > **Part II**
> >
> > 9. On the first version of the paper, we had four baselines (Unconditional, Predictor-guided, PepCVAE and Greener et al.). The three baselines Unconditional, Predictor-guided and Greener et al. were trained on the full sequences up to 2048 amino acids. PepCVAE was trained on the first 32 amino acids.
> > Regarding this last point, of course training an RNN on 2048 tokens is not an issue. However, retraining a functional VAE model in combination with RNNs is. This type of models have known issues with posterior collapse (see e.g. Bowman et al. 2016) which is also why we did not choose these models for our own approach. We were not able to train a stable model with full sequence length without significant changes to the architecture the authors of PepCVAE proposed, which was trained originally on shorter (L=30) peptide sequences. We moved the results of PepCVAE and ProteoGAN trained on sequences of length 32 amino-acids to the supplementary material (Figure A2) as they are not directly addressing the problem at stake.
> > All other baselines were trained on the original sequences up to 2048 amino acids.
> > 10. The conditioning was another point that was unclear among several reviewers, and we have clarified the structure of the data and how our model incorporates the label data. In short, our model allows for conditioning on multiple GO terms simultaneously (all the GO labels of a given protein) by summing the respective label embeddings into a final vector that is fed to the cGAN. This allows to leverage the hierarchical structure of the labels efficiently.
> > 11. Thank you for pointing out this relevant reference, we have added it in our revised manuscript.
> >
> > Things that would improve my rating: [...]
> >
> > We believe your comments enriched our contribution significantly. In addition, we have (1) revisited our train/val/splits and better described them in the main text, (2) added several baselines and explained why MMD is a good measure of sequence generative quality, and (3) updated the manuscript to make it clearer to readers. We therefore hope we could address your concerns and that this will increase the value of our work in your eyes. Please let us know if there are any more open questions.

---

### Author Response · Authors · 2020-11-23
**General answer to all reviewers**

We warmly thank all the reviewers for their thoughtful feedback on our work. The rebuttal is composed of a global answer, which addresses questions asked by several reviewers, specific answers to each reviewer and the update of the main document and supplementary material, with edits marked in color.

Several reviewers questioned the train/validation/test splits that we used in the paper to compare the performance of our model to baselines. First, we would like to acknowledge that keeping 4% (2% + 2%) of the sequences for the validation and test sets was unusually small and the results were potentially subject to having a large variance. Therefore, we reran our model and the baselines (including new ones) keeping 10% of the sequences as validation set and 10% of the sequences as test set. Additionally, we trained and tested the model and baselines on 5 different splits to obtain a standard deviation for each model and criterion. Second, the current splits (80-10-10) are done uniformly at random. We are doing so because our objective is to show that it is possible to generate sequences that verify the following four points:
1) Novel sequences, defined as sequences not identical to the ones in the training set. We show, in this revision, that the generated sequences of our model are different from the sequences in the training set, and that the distribution of pairwise distances, in the kernel feature space, between generated sequences and its nearest neighbour in the training set are close to the distribution of pairwise distances between test set sequences and its nearest neighbour among training set sequences (compare Figure A8).
2) In-distribution sequences, namely sequences that follow the distribution of sequences in the dataset. We show this by computing the MMD score between the distributions of generated sequences and real sequences. As a side remark, the results are robust to the choice of the kernel used for MMD, whether we choose a linear or a gaussian kernel on the k-mer embeddings (see Table 1.).
3) In-distribution sequences conditionally generated based on given GO labels. We verify this by measuring MRR.
4) Sequences that show the same physico-chemical characteristics as real sequences, as evaluated by ProFET. This last evaluation is a solid first step towards checking whether the generated sequences could behave as real sequences if inserted in a biological environment.
	We believe that being able to generate sequences that follow these four criteria constitutes a big step forward compared to the state-of-the-art. However, we agree that being able to generate out-of-distribution sequences with splits that are pre-filtered using CATH or SCOP would be the next frontier in this line of work, and hope some of our results will prove to be useful for the community when moving towards that goal.

Several reviewers also questioned the meaning of the word “valid”. We used the adjective “valid” to describe generated sequences that are in-distribution with sequences in the training set (i.e. exhibiting a low MMD). We agree that this word is not precisely used and propose instead to describe such sequences as “validated in-silico”. This would describe generated sequences that are verifying the four points described above. Therefore we consider that such sequences are plausibly viable, i.e. plausibly exhibiting the given function when inserted in a cell. This would further need to be confirmed by biological experiments in a future work.

Overall, we believe that the suggestions from the reviewers strongly improve the paper and we look forward to hearing their impressions on this revised version.

---

### Decision · Program_Chairs · 2021-01-07
**Final Decision**

**Decision:**

Reject

**Comment:**

The paper proposes to use conditional GANs to generate protein sequence with respect to GO molecular functions. The idea is nice. But the reviewers find that there are many things that are not clear. For example, some sentences, phrase, the model and experiments pointed by the reviewers that should be  rigorous described. The technical contribution is also limited. The author are encouraged to revise the paper according to the comments.